# DNA replication initiation drives focal mutagenesis and rearrangements in human cancers

Pierre Murat [1,2] ✉, Guillaume Guilbaud [1] & Julian E. Sale [1] ✉

The rate and pattern of mutagenesis in cancer genomes is significantly influenced by DNA accessibility and active biological processes. Here we show that efficient sites of replication initiation drive and modulate specific mutational processes in cancer. Sites of replication initiation impede nucleotide excision repair in melanoma and are off-targets for activation-induced deaminase (AICDA) activity in lymphomas. Using ductal pancreatic adenocarcinoma as a cancer model, we demonstrate that the initiation of DNA synthesis is error-prone at G-quadruplex-forming sequences in tumours displaying markers of replication stress, resulting in a previously recognised but uncharacterised mutational signature. Finally, we demonstrate that replication origins serve as hotspots for genomic rearrangements, including structural and copy number variations. These findings reveal replication origins as functional determinants of tumour biology and demonstrate that replication initiation both passively and actively drives focal mutagenesis in cancer genomes.

Cancer cells are characterised by their ability to sustain cell growth and proliferation by evading suppressor signals and activating oncogene expression[1]. The overexpression of oncogenes such as c-Myc, h-RAS, or Cyclin E shortens the G1 phase of the cell cycle, expedites S phase entry, and increases levels of replication initiation[2–4]. This accelerated S-phase entry is linked to replication stress and genome instability, in part due to origin over-usage and activation of dormant replication origins[5]. However, the impact of replication initiation on mutagenesis in human cancers has not been examined.

We recently reported that a subset of highly efficient replication origins, conserved across a range of untransformed, immortalised and transformed human cell lines[6], act as hotspots for mutagenesis in the germline[7]. These 'constitutive' replication origins are notably enriched within the active and early replicated regions of the human genome, particularly within gene bodies. The mutational processes operating at these sites thus contribute to elevated mutational burdens at gene promoters and splice junctions, impacting phenotypic diversity across human populations. Genomic regions in which replication initiates constitute fragile sites associated with DNA breaks and translocations in cancer[4,8], and origin activity has also been shown to drive

translocations at the *MYC* locus in murine B cells, suggesting a potential link between replication origins and the development of lymphomas and other cancers[9]. Further, increased origin firing contributes to replication stress driven by oncogene activation and is linked to DNA breaks and rearrangements[4] while overexpression of origin firing factors such as GINS1 and CDC45 in human colorectal cancer cells leads to aneuploidy[10]. Collectively, these studies highlight replication origin firing as a cancer-relevant trigger for genome instability.

In this study, we report the identification and characterisation of mutational processes occurring at constitutive origins in cancer. Our results emphasise the major impact of replication initiation in shaping the mutational landscape of the human genome, adding replication origins to the expanding repertoire of genomic features influencing local mutation rates in cancer (reviewed in ref. 11).

## Results

### Pan-cancer analysis of mutation distribution reveals enrichment at replication origins

To systematically examine the frequency of somatic point mutations at constitutive origins, we considered mutation calls from the

[1]Division of Protein & Nucleic Acid Chemistry, MRC Laboratory of Molecular Biology, Francis Crick Avenue, Cambridge CB2 0QH, UK. [2]Present address: Wellcome Sanger Institute, Hinxton CB10 1RQ, UK. ✉e-mail: pm23@sanger.ac.uk; jes@mrc-lmb.cam.ac.uk

International Cancer Genome Consortium (ICGC) aggregated from 86 cancer projects. To evaluate a potential enrichment of single nucleotide variants, hereafter refer to as mutations, at constitutive origins compared to baseline levels, we first examined 'isolated' origins (refer to the Methods section for detailed information on the analysed origin sites) and found a ~1.7-fold enrichment at these sites compared to their flanks (Supplementary Fig. 1a). This was accompanied by an increase in minor allele frequencies (Supplementary Fig. 1b). We confirmed that constitutive origins located within origin clusters exhibit a similar mutational burden (Supplementary Fig. 1c, d) supporting the use of 'isolated' origins to avoid potential biases introduced by clustering when estimating local changes in mutation rates. To account for local variation in base composition, we corrected the apparent mutation rates of each of the six pyrimidine mutations (and hence also reciprocally mutation rates at purines) by their occurrences and found increased mutation rates associated to each type (Fig. 1a). When adjusted for trinucleotide frequency, we observed increased mutation rates for all mutations except C > T (Supplementary Fig. 1e). However, we found that C > T transitions are favoured in a non-CpG context (Supplementary Fig. 1f), suggesting that constitutive origins protect against spontaneous CpG deamination, as observed in the germline[7]. These observations suggest that a specific form, or forms, of mutagenesis is focused on constitutive origins in cancer genomes. We note that mutation rates are negligible at 'core' origins, a class of cell type-independent origins (see the Methods section for detailed information)[12], which do not overlap with constitutive origins,

(Supplementary Fig. 1g). This suggests that constitutive origins represent the most recurrent sites of replication initiation in cancer genomes, accumulating a significant mutational imprint.

We then asked whether mutation rates at constitutive origins vary across cancer types. We computed origin mutation rates for individual tumours with over 5000 mutations (1056 tumours across 20 cancer types) to ensure reliable estimates of mutation density at and near origins. While some cancers, like gall bladder cancer, show high mutation rates at origins (Fig. 1b), others, like renal or liver cancers, unexpectedly have constant or even negative rates. To assess variation within cancer types, we computed the ratios of mutations at origins compared to flanking regions and assessed their variability across cancer types (Fig. 1c). This revealed substantial differences within specific cancer types, pancreatic cancer, for instance, showing ratios ranging from 0.21 to 5.14. These observations suggest that increased mutation rates at origins are driven by mutational processes of varying degrees of intensity rather than inherent tumour properties. Supporting this, we found a strong correlation between the total number of mutations and those at origins (*Rho* = 0.954, Supplementary Fig. 1h), but an opposite correlation with the origin/flank mutation ratio (*Rho* = −0.246, Supplementary Fig. 1i). This pattern suggests that specific mutational processes operate at constitutive origins.

To determine whether these mutational processes are cancer-specific or simply reflect origin usage in somatic tissues, we investigated mutation rates at constitutive origins in normal tissues. First, we examined mutation rates at origins in non-dividing cells across various tissues[13] and found a mutational burden at constitutive origins similar to that of their flanks (Supplementary Fig. 2a), with no significant increase in mutation rates for any of the six pyrimidine mutations (Supplementary Fig. 2b). This suggests that mutagenesis at origins does not occur in the absence of replication. We then examined the distribution of mutations at constitutive origins in 18 somatic cell types[14] and observed an accumulation of SNVs at these sites (Supplementary Fig. 2c). However, after adjusting for local variations in base composition, the mutation rates for the six pyrimidines did not rise above background levels (Supplementary Fig. 2d, e). This suggests that the apparent accumulation of mutations at origins in non-cancerous replicating tissues reflects local variation in base composition rather than active mutational processes. Overall, these findings support the existence of cancer-specific mutational processes targeting constitutive origins.

## Origin-associated mutational signatures
To identify and characterise these mutational processes, we clustered cancer samples based on the cosine similarity of their origin trinucleotide mutation signature (Fig. 2a). We chose a clustering approach over the application of non-negative matrix factorisation, as it enables the grouping of mutational patterns and exposures, facilitating the identification of mutational patterns that are present genome-wide but more prevalent at replication origins. Our analysis specifically focused on tumours with over 50 mutations at origins (227 tumours across 17 cancer types) to ensure sufficient power to call distinct signatures. Each origin signature underwent adjustment for local trinucleotide composition and correction to account for background values from the flanking domains (details in Methods). The clustering revealed five distinct tumour groups, with two clusters exclusive to a specific cancer type and three including multiple types (Fig. 2b). Subsequently, representative mutation signatures were computed by aggregating mutations at origins from tumours within each cluster (Fig. 2c). These were compared to known signatures of somatic mutations in cancerous human tissues (COSMIC SBS signatures, Supplementary Fig. 3a)[15]. Further deconvolution of cluster 5, which visually appeared to contain several potential subclusters, did not reveal meaningful tumour groups (Supplementary Fig. 3b, c).

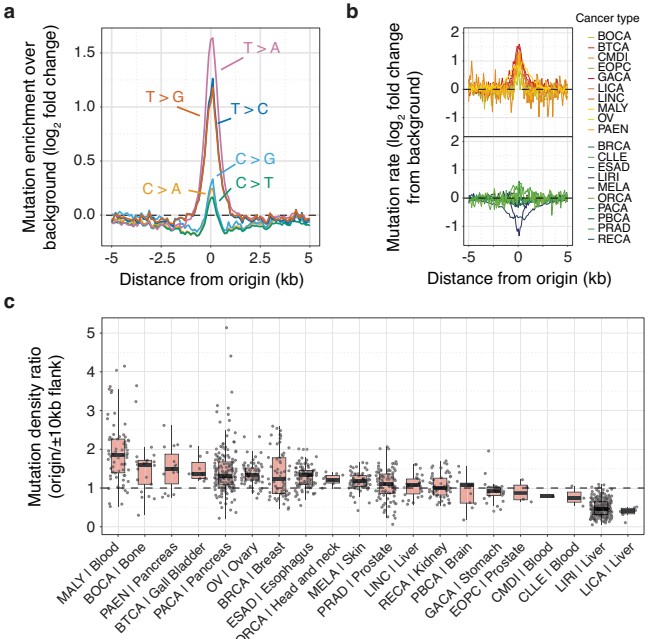

**Fig. 1 | Pan-cancer analysis of mutation distribution across the genome reveals enrichment at constitutive origins. a** Mutation rates associated with the six pyrimidine substitutions at constitutive origins. Mutation rates were computed from aggregated mutation calls from 86 ICGC cancer projects, corrected for local variation in base composition and background values. **b** Averaged and background adjusted mutation rates stratified by cancer types. Acronyms of cancer types are detailed in the Methods section. **c** Distribution of origins / origin flanks mutation density ratios of individual cancer samples stratified by cancer type and primary sites. Density ratios were computed by considering origin domains (origin midpoints ± 500 bp) and origin flanking domains (origin midpoints ± 10 kb excluding origin domains). Box plots show medians and interquartile ranges. Individual cancer samples are shown as grey dots. Only whole-genome-sequenced cancer samples with at least 5,000 mutations were considered for experiments reported in (**b**, **c**).

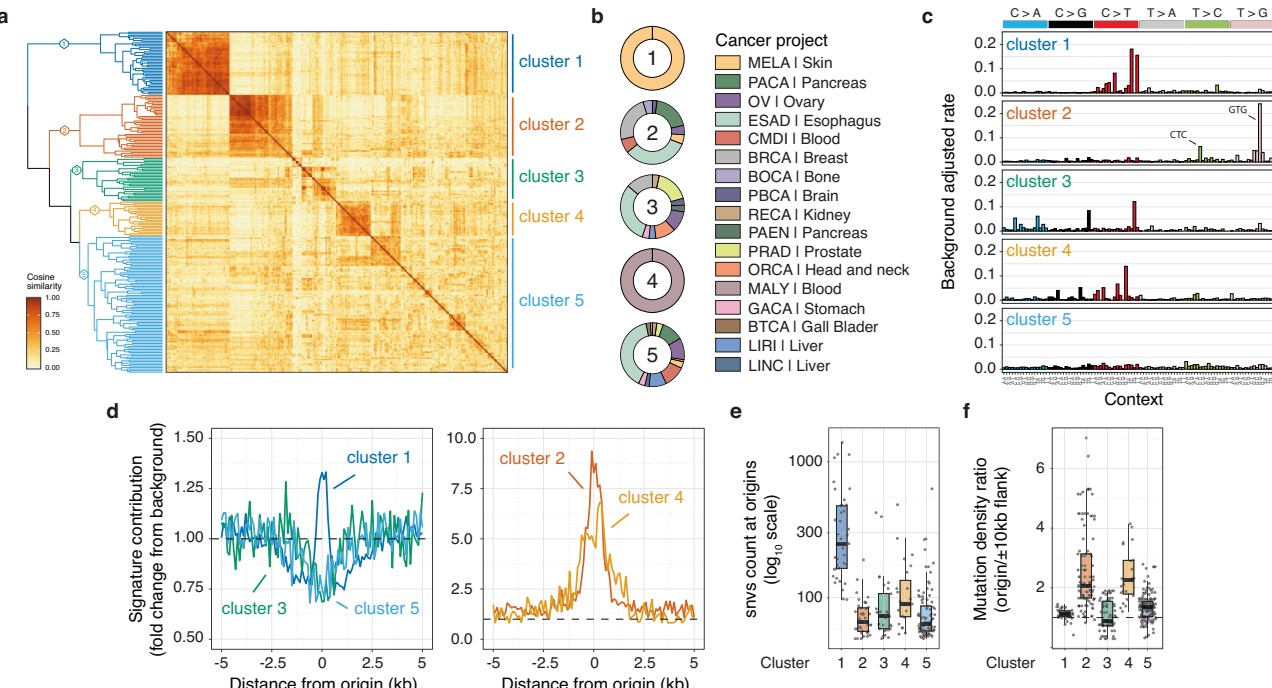

**Fig. 2 | Signature analysis uncovers distinct mutagenic processes focused at constitutive origins. a** Cancer sample clustering based on the cosine similarity of their origin trinucleotide mutational signature. Each signature at origin domains was corrected to adjust for local trinucleotide composition and background values from neighbouring origin flanking domains. Only whole-genome-sequenced cancer samples with at least 50 mutations at origins were considered. **b** Distribution of cancer types and primary sites within the five identified clusters of tumours. **c** Origin-associated mutational signatures for tumour clusters. Signatures were computed from aggregated mutation calls at origin domains for clustered cancer samples and corrected as previously. **d** Local exposure to mutational signatures associated with clustered tumours. Signature contributions are corrected for background origin flank values. **e** Total mutation count at origins and (**f**) origin/ origin flanks mutation density ratios for individual cancer samples grouped by clusters. Box plot show medians and interquartile ranges. Individual cancer samples are represented as grey dots.

Clusters 1 and 4 are specific to skin melanomas (MELA) and B cell malignant lymphomas (MALY) respectively, displaying characteristic origin signatures highly similar to the known signatures SBS7b (cosine similarity = 0.83, exposure to ultraviolet light) and SBS84 (cosine similarity = 0.88, activity of activation-induced cytidine deaminase (AICDA or AID)). Cluster 2 encompasses tumours from various cancer types, with pancreatic ductal adenocarcinoma (PACA), oesophageal adenocarcinoma (ESAD), and breast cancer (BRCA) being the primary contributors. The cluster 2 signature exhibits high similarity to SBS43 (cosine similarity = 0.95), a signature of unknown aetiology, marked by T > G mutations in the specific context GTG. Clusters 3 and 5 comprise samples from over 10 different cancer types. The cluster 3 signature does not share similarities with any reported COSMIC signatures. Lastly, the aggregated cluster 5 signature appears featureless, resembling a 'flat' signature probably resulting from merging signals originating from diverse processes. It is noteworthy that clusters 3 and 5 are the least resolved clusters in our analysis (Fig. 2a). We then compared these origin-associated signatures with genome-wide signatures derived from samples within the respective clusters (Supplementary Fig. 3d). Aside from cluster 1, our origin-associated signatures exhibit minimal resemblance to the corresponding genome-wide signatures. This finding validates our approach for identifying processes that focus mutagenesis at origins.

To gain a deeper understanding of the specificity of our origin-associated signatures, we examined their local contribution near origins (Fig. 2d). The aggregate origin signatures derived for clusters 3 and 5 show no discernible contribution to origin mutagenesis. Conversely, processes linked to cluster 2 and 4 signatures focus mutations at origins, with a substantial 7- to 9-fold increase compared to origin flanks. The process related to the cluster 1 signature exhibits some preference for origins, albeit with a moderate 1.3-fold enrichment. We further quantified the number of mutations attributed to each signature at origins (Fig. 2e) and evaluated the origin specificity of associated processes (Fig. 2f) at the level of individual cancer samples. These analyses reveal that while the process at origins within cluster 1 lacks specificity for origins, it is enriched at origins and contributes significantly to origin mutagenesis in the MELA project samples. Conversely, clusters 2 and 4 are highly specific to origins but contribute only moderately in terms of absolute mutation numbers. We validated these findings by fitting COSMIC SBS signatures to mutational profiles of origin domains (Supplementary Fig. 3e). Overall, our analysis identifies three independent mutational processes targeting mutagenesis at constitutive origins, two of which are tissue-specific, while one operates across different cancer types. Next, our aim was to identify the aetiology of these processes, beginning with those specific to particular tissues.

## Differential DNA repair underlies mutation hotspots at origins in skin melanoma

Samples from cluster 1 exclusively originate from skin melanomas and exhibit both origin and genome-wide signatures that closely resembles the known COSMIC SBS7b signature (Fig. 2b, c and Supplementary Fig. 3a), which is linked to UV light exposure. However, the absolute contribution of SBS7b is ~2.5-fold higher at origins compared to their flanks (Supplementary Fig. 3e). This could be explained by either locally impaired DNA repair or a higher probability of UV-induced lesions at constitutive origins. Given that both protein binding and transcription initiation have been reported to impair nucleotide excision repair (NER)[16,17], we hypothesised that increased mutagenesis at origins may, by analogy, result from local NER deficiency due to the loading of the pre-replication complex at licensed origins.

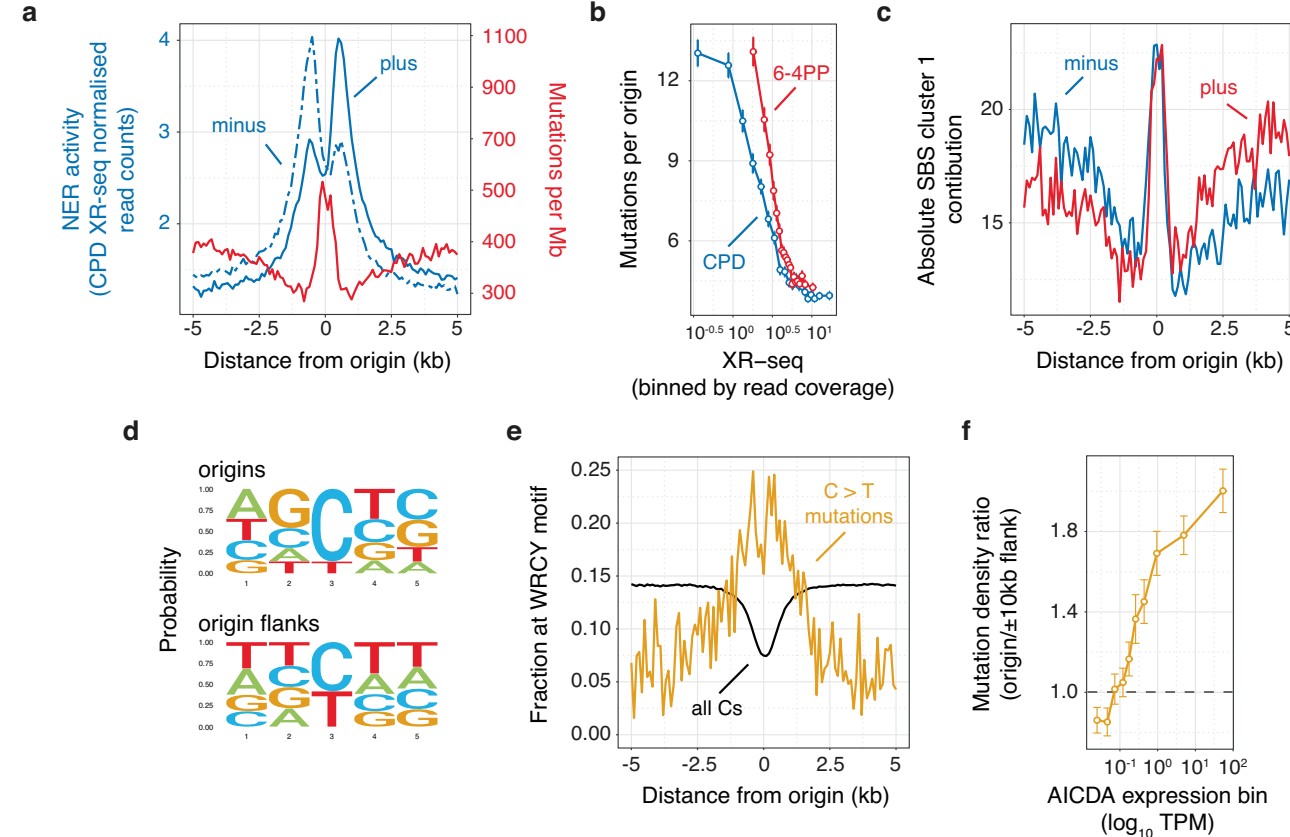

**Fig. 3 | Differential DNA repair and off-target AID deamination underlie mutation hotspots at origins.** Nucleotide excision repair in ultraviolet-irradiated human cells inversely mirrors mutation density at origins in skin melanomas. **a** Average melanoma mutation density for cluster 1 tumour samples (red line) and strand-resolved XR-seq profiles for CPD (blue lines) in CSB/ERCC6 mutant NHF1 skin fibroblasts at constitutive origins. **b** Number of mutations at origins as a function of XR-seq signals for CPD (blue line) or 6–4 PP (red line). XR-seq signals were binned by read coverage. Mutation data represent the means and standard errors to the mean of values for each XR-seq signal bin. **c** Strand-resolved cluster 1 mutational signature contribution to mutagenesis at the origins of cluster 1 tumours. Signature contribution was computed by considering aggregated mutation calls from cluster 1 tumours. Constitutive origins are off-targets for AID deamination in malignant B cell lymphomas. **d** Consensus contexts of mutations mapped within origin or origin flank domains for cluster 4 tumours. The central position represents the mutated bases. **e** Fraction of C > T mutations (orange line) or C residues (black line) overlapping the WRCY AID hotspot motif (where W represents weak bases, R purines, C the mutated bases and Y pyrimidines) at origins of cluster 4 tumours. **f** Pan-cancer origins/origin flanks mutation density ratios as a function of AID (AICDA) expression. Mutation density ratio values were computed for transcript per million (TPM) bins and represent the means and standard errors of the mean.

To assess this point, we used genome-wide maps of NER activity, obtained through XR-seq analysis of cyclobutane pyrimidine dimers (CPD) and pyrimidine-pyrimidone (6–4) photoproducts (6–4 PP) in irradiated skin fibroblasts[18]. We correlated the XR-seq profiles with DNase I hypersensitivity (DHS) at origins (Supplementary Fig. 4a) and observed a positive correlation between NER activity and origin accessibility ($Rho = 0.40$ and $0.17$ for CPD and 6–4 PP respectively, Supplementary Fig. 4b) suggesting increased accessibility is associated with higher levels of damage. However, we found that the profile of mutation density at origins inversely mirrors NER activity (Fig. 3a and Supplementary Fig. 4c) and that the number of mutations at origins displayed a negative correlation with XR-seq signals at these sites ($Rho = -0.44$ and $-0.36$ for CPD and 6–4 PP respectively, Fig. 3b). Thus, replication initiation may specifically impair NER explaining the increased in SBS7b mutagenesis at origins in melanoma.

To control for other known processes that may impair NER at constitutive origins, such as the binding of transcription factors (Supplementary Fig. 4d, e), ref. 16 we compared NER activity at origins with or without experimentally validated binding sites of transcription factors, such as CTCF or ETS1. We found that the local NER deficiency observed at origins does not depend on transcription factor occupancy at these sites (Supplementary Fig. 4f). It worth noting that we performed this analysis using data from CSB/ERCC6 mutants to interrogate global NER rather transcription-coupled NER. We further ruled out the involvement of transcription-coupled NER at origins by observing that XR-seq profiles at intergenic origins were comparable to those observed for origins located at promoters or within gene bodies (Supplementary Fig. 3g).

Interestingly, when examining strand-resolved XR-seq signals (Fig. 3a and Supplementary Fig. 4c), we observed a preference for NER activity on the lagging strand near constitutive origins. To determine whether this observation arises from a replication strand bias or from a skew in UV-reactive dinucleotides (such as TT dinucleotides), we adjusted XR-seq signals for dinucleotide composition skew (details in Methods) and calculated strand biases (Supplementary Fig. 3h). Despite the correction, NER activity still exhibited a bias toward the lagging strand. Subsequently, we evaluated the strand specificity of our cluster 1 origin-associated signature (Fig. 3c) and found no strand bias at the origins but an enhanced contribution on the lagging strand in close proximity to the origin. Taken together, these findings support NER deficiency at origins and increased NER on the lagging strand of replication forks translocating away from origins.

## Origins are off-targets of cytosine deaminases in B cell lymphomas

The cluster 4 signature (Fig. 2c) exhibits notable specificity for origins (Supplementary Fig. 2b) and bears significant similarity to the established COSMIC SBS84 signature associated with AID activity in lymphoid cancers. We hypothesised that origins might serve as off-targets for AID deamination. To investigate this point, we first examined the extended context of cluster 4 mutations at origins or their flanks (Supplementary Fig. 4i). Our analysis revealed that C > T mutations at origins occur within specific extended contexts, distinct from those observed at origin flanks, characterised by an enrichment of As, Gs, and Cs at the −2, −1, and +1 positions respectively relative to the C > T mutations. This mutation context closely resembles the known WRCY AID hotspot motif (where W represents weak bases, R purines, C the mutated base and Y pyrimidines)[19,20]. Subsequently, by constructing consensus sequences for C > T mutations at origins and origin flanks (Fig. 3d), we confirmed that mutagenesis at origins in Cluster 4 genomes aligns with this consensus sequence. Furthermore, to demonstrate the activity of AID at origins in lymphomas, we calculated the fraction of C > T mutations within the canonical WRCY motif around origins (Fig. 3e) and observed a local increase at origins. Interestingly, the fraction of Cs in a similar context decreased at origins, supporting that origins are indeed off-targets of AID. To reinforce this observation, we identified a pan-cancer correlation between mutational burden at origins and AID expression (*Rho* = 0.25, Fig. 3f).

Another source of deamination in lymphomas and other cancer genomes arises from the activity of APOBEC deaminases[21,22]. We therefore sought to determine whether APOBEC activity contributes to mutagenesis at origins. To investigate this, we initially examined whether high expression of APOBEC in lymphomas is associated with increased mutational burden at origins (Supplementary Fig. 4j). Our analysis revealed that the overexpression of APOBEC3 enzymes is indeed associated with higher origin/flank mutation density ratios, suggesting a potential contribution of APOBEC3 to origin mutagenesis. However, when assessing the background-adjusted origin mutation profiles computed for lymphoma genomes, we observed minimal contribution from the COSMIC SBS2 and SBS13 signatures, which are attributed to the activity of the APOBEC family of cytidine deaminases (Supplementary Fig. 4k). This observation suggests that cytosine deamination at origins in B cell germinal centre lymphomas is primarily driven by AID.

## Polymerase δ behaviour at non-B DNA structures drives origin mutagenesis

Cluster 2 samples are derived from a wide range of cancer types, with focal mutagenesis at origins primarily driven by T > G mutations (see Fig. 2c and Supplementary Fig. 5a). We note that this signature is independently observed across all cancer types within cluster 2 tumours (Supplementary Fig. 5b), suggesting a common mechanism that is not tissue-specific. Investigating this distinctive signature, we first examined the extended contexts of these mutations at origins and their flanks (Fig. 4a). Interestingly, T > G mutations at origins, unlike those within origin flanks, tend to occur within G-rich sequences, particularly embedded within G tracts. This suggests a potential association with non-B DNA structures, such as G-quadruplexes (G4s), known to hinder replication and induce mutagenesis and genetic instability[23,24]. To probe the role of G4s in mutagenesis at origins, we analysed short sequences overlapping with T > G mutations at origins and their flanks (25 bp around mutations), predicting their susceptibility to fold into G4 structures using the G4Hunter algorithm, which considers G-richness and G-skewness to generate a quadruplex propensity score (G4H score)[25]. Our analysis revealed that sequences surrounding T > G mutations at origins in cluster 2 genomes exhibit significantly higher overall positive G4H scores compared to sequences containing T > G mutations in origin flanks or sequences

containing any T residues at origins (Fig. 4b). This indicates a potential driving role of G4 structures in mutagenesis at origins. We next identified sequences with the potential for G4 formation following a $N_5G_{3+}N_{1-12}G_{3+}N_{1-12}G_{3+}N_{1-12}G_{3+}N_5$ pattern (where N is any base)[26] within origin domains and computed their G4H scores. Remarkably, sequences bearing T > G mutations display higher G4H scores than those without mutations (Fig. 4c), suggesting that G4 structures, rather than solely G-rich sequences, are driving mutagenesis, with the most stable predicted G4s being the most mutated. We confirmed this by examining mutagenesis at G4 structures experimentally mapped in vivo using data generated by the G4access method[27]. Experimentally validated G4 structures mutated at origins are predicted to be more stable (Supplementary Fig. 5c) and have shorter loops (Supplementary Fig. 5d, e) than those that are not mutated.

Having established the role of G4s in driving mutagenesis at origins, we delved deeper into understanding the characteristics of our cluster 2 origin signature. We hypothesised that T > G mutations in the specific GTG context (Fig. 2c) might be disproportionately represented in G4-forming sequences. We analysed the occurrence of each of the 96 trinucleotide contexts within the established G4 origin pattern and revealed an enrichment of the GTG context within G4 forming sequences at origins (Supplementary Fig. 5f). Accounting for the trinucleotide composition of G4s enabled us to extract a mutational signature specific to G4s (Supplementary Fig. 5g), illustrating the diverse spectrum of mutations observed at these sites. In this context, we note that G4s contribute on average ~33%, and up to 72%, to mutagenesis at the origins of cluster 2 tumours (Supplementary Fig. 5h, i).

Furthermore, we examined the mutation pattern at origin G4s by precisely mapping mutations at G4 sites oriented by the direction of replication (Fig. 4d). Our findings indicate an accumulation of mutations at the first nucleotides before and after the initial G tract of G4s, aligning with biochemical observations[28,29], suggesting that polymerases may stall or proceed with errors past the first G tract of G4s during polymerase extension. This supports that mutagenesis at origins in cluster 2 is primarily a result of polymerase stalling at G4s. However, we observed that our cluster 2 signature does not exhibit a replication strand bias at origins (Supplementary Fig. 5j), and that G4s at origins are not predicted to be inherently more stable than G4s at their flanks (Supplementary Fig. 5k). This suggests that a single replicative polymerase may be responsible for mutagenesis at origins, and its synthesis errors are confined within origin domains.

Evidence, from both biochemical studies in yeast and genetic investigations in humans, supports the theory that following the priming of DNA replication by polymerase α, polymerase δ initiates both leading and lagging strand synthesis, thus performing the bulk of DNA synthesis near replication origins[7,30–32]. This initial DNA synthesis following origin firing occurs uncoupled from helicase activity and is likely more susceptible to polymerase stalling, potentially accounting for the increased mutation rate observed at G4s near origins. Consequently, we evaluated the exposure of origins to mutational signatures associated with the activity of POLD1 and POLE, which become more pronounced when the exonuclease domains of these polymerases are compromised[33]. Our analysis revealed that the local distribution of these signatures aligns with polymerase δ being the active polymerase within 1-kilobase domains centred on human origins (Fig. 4e). Taken together, these findings support a model wherein polymerase δ stalls and triggers error-prone DNA synthesis at G4 structures in the immediate vicinity of replication origins of cluster 2 tumours.

## Mutagenesis at origins in pancreatic adenocarcinoma (PACA) correlates with markers of increased proliferative drive and replicative stress

DNA secondary structure-dependent mutagenesis at origins is shared among various cancer types, although it is not uniformly observed

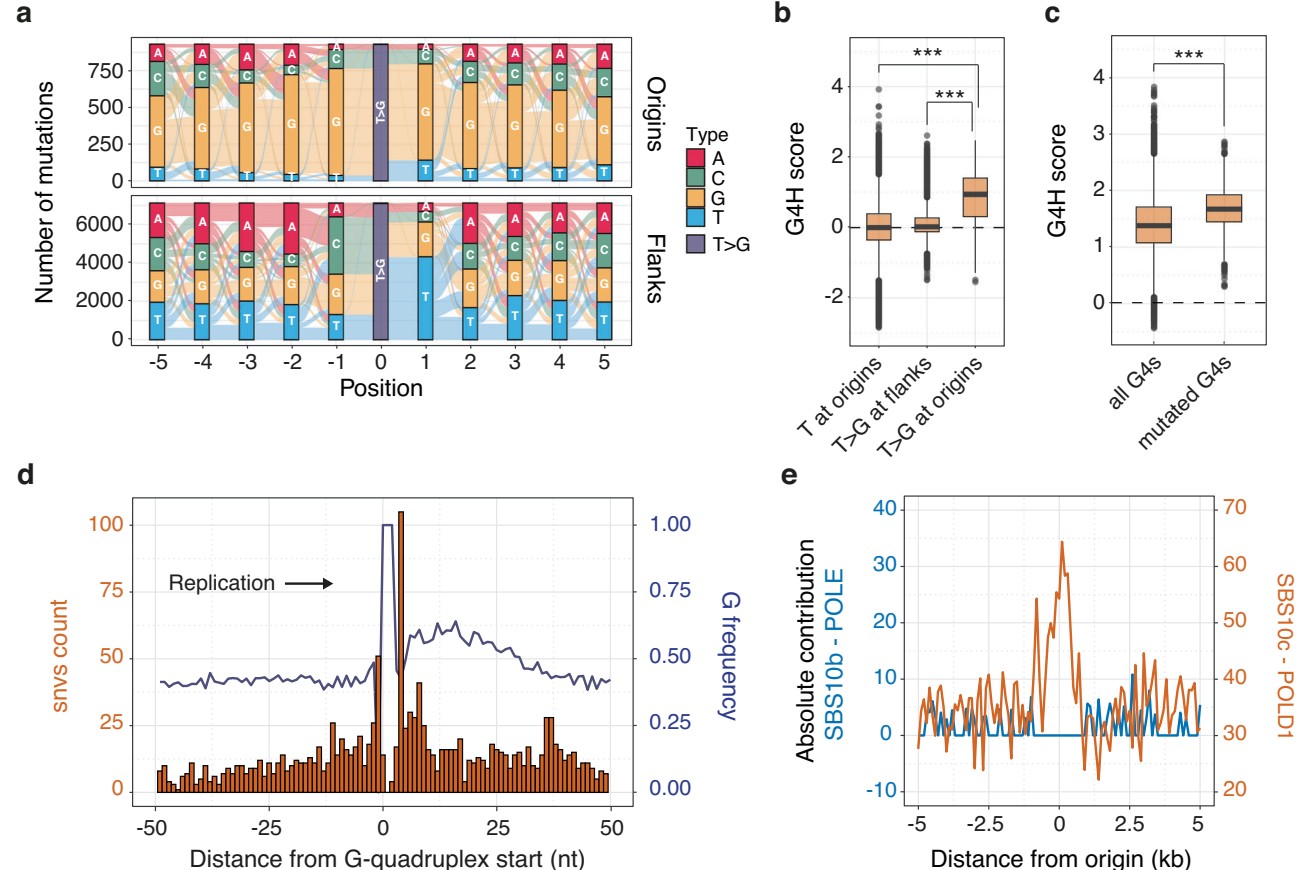

**Fig. 4 | Polymerase δ behaviour at G-quadruplex structures drives origin mutagenesis in diverse cancer types. a** River plot illustrating the sequence contexts of T > G mutations at origins or origin flanks for cluster 2 tumour samples. **b** Distribution of the G-quadruplex (G4) propensity scores, G4H scores, associated with short 51 bp sequences encompassing T > G substitutions at origins and origin flanks, or any T residues at origins. **c** Distribution of G4H scores for all or T > G mutated sequences at origin domains, fitting the pattern $N_5G_{3+}N_{1–12}G_{3+}N_{1–12}G_{3+}$ $N_{1–12}G_{3+}N_5$ (where N is any base), representing potential G4 formation. Box plot report medians and interquartile ranges. Outliers are shown as grey dots.

***$P < 0.001$, Kolmogorov-Smirnov test. **d** Distribution at single-nucleotide resolution of mutations and G frequency at G4-forming sequences found within 1 kb domains centred on origins. The start of the G4s is defined as the first G from the first G tract of the previous pattern. G4-forming sequences were oriented toward the direction of replication. **e** Absolute contribution of COSMIC signatures SBS10b (blue, left hand Y-axis) and SBS10c (red, right hand Y-axis) to mutagenesis at origins, associated with the activity of POLE and POLD1, respectively. Both signatures are attributed to defective proofreading due to acquired mutations in the exonuclease domains of the polymerases.

within all samples of the same cancer type. Our next goal was to elucidate the specific cancer features that influence the intensity of mutagenesis at origins. We selected pancreatic ductal adenocarcinoma (PACA) as a model cancer due to its wide range of variation in the ratio of mutations at origins to flanks (Fig. 1c) and the availability of a sufficiently comprehensive matched transcriptomic dataset. We first identified distinct cancer subtypes within PACA based on transcriptomic profiles. Using a uniform manifold approximation and projection dimensionality reduction (UMAP) on normalised gene expression counts of individual samples (Fig. 5a), we identified three subtypes, henceforth referred to as PACA 1 to 3, each exhibiting different levels of mutagenesis at origins. While all subtypes show similar genome-wide mutation burdens (Supplementary Fig. 6a), PACA 1 exhibits a higher degree of mutagenesis at origins compared to PACA 2 and 3 (Fig. 5b). Notably, tumours classified as PACA 1 were characterised by the presence of our cluster 2 origin signature, a feature attenuated in PACA 2 and absent in PACA 3 (Supplementary Fig. 6b), suggesting that DNA structure-dependent mutagenesis at origins occurs to varying extents in these cancer subtypes. Consistently, exposure to our cluster 2 signature at origins was higher in PACA 1 compared to PACA 2 and 3 (Fig. 5c). These observations indicate that UMAP reduction enables the stratification of pancreatic cancers into

distinct subtypes, facilitating the study of determinants influencing mutagenesis at origins.

We proceeded with a differential gene expression analysis to identify genes displaying altered expression in PACA 1 compared to PACA 2 and 3 (Supplementary Fig. 6c). Subsequently, the results of this analysis were used for a gene set enrichment analysis (GSEA) to explore the altered KEGG pathways in PACA 1 (Fig. 5d and Supplementary Fig. 6d, details in Methods). The GSEA revealed the upregulation of cancer pathways ($P = 7.44 × 10^{-5}$, permutation test) linked with increased cell growth and proliferation, such as the JAK-STAT or MAPK pathways[34,35]. This led us to hypothesise that increased mutation rates at origins might stem directly from replicative stress induced by excessive cell proliferation. To address this hypothesis, we evaluated the expression of biomarkers associated with replicative stress in pancreatic cancers[36] and discovered that tumours from PACA 1 and PACA 2 subtypes indeed exhibit cellular stresses interfering with DNA replication (Fig. 5e), contrasting with the absence of such stresses in PACA 3. However, PACA 1 and PACA 2 tumours displayed similar levels of replicative stress biomarkers, suggesting other contributing factors. We speculated that these subtypes might diverge in their growth profiles. To investigate this further, we inferred the duration and respective phases of their cell cycles from the normalised expression

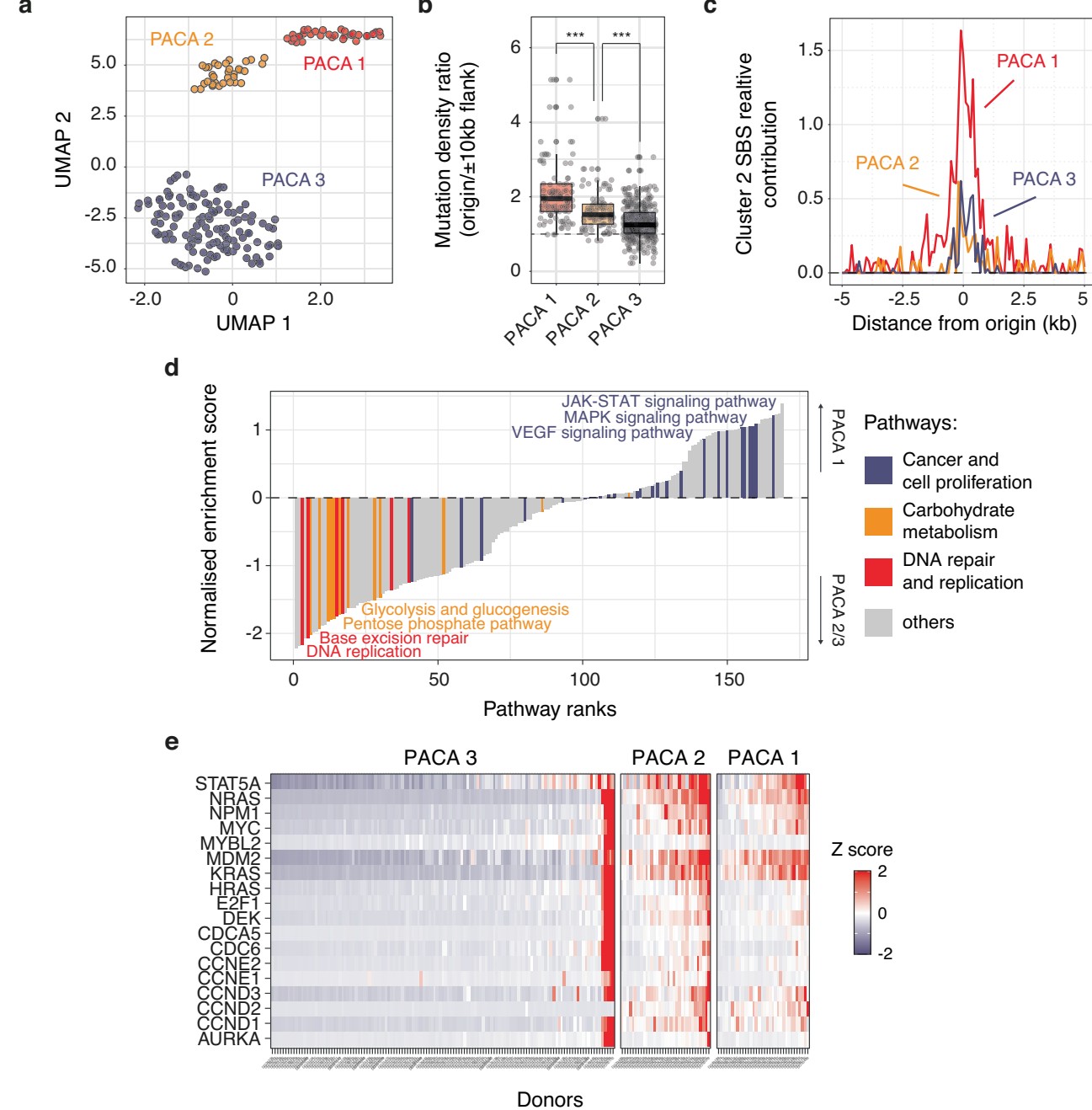

**Fig. 5 | Replicative stress exacerbates origin mutagenesis in pancreatic ductal adenocarcinoma. a** UMAP dimensionality reduction of transcriptomic profiles from 194 pancreatic ductal adenocarcinoma samples (PACA project) reveals three cancer subtypes denoted PACA 1 to 3. **b** Distribution of origins/origin flanks mutation density ratios for individual cancer samples categorised by PACA subtypes. Box plot report medians and interquartile ranges. Grey dots represent individual cancer samples. ***$P < 0.001$, Kolmogorov-Smirnov test. **c** Relative contribution of Cluster 2 signature to mutagenesis at origins. **d** Pathway enrichment analysis of genes dysregulated in PACA subtype 1 compared to subtypes 2 and 3. Normalised enrichment scores indicate the extent of upregulation of gene sets representing KEGG pathways in PACA 1 tumours. Selected dysregulated pathways are colour-coded: cancer and cell proliferation (blue), carbohydrate metabolism (orange), and DNA repair and replication (red). **e** Gene expression levels of replication stress biomarkers in pancreatic cancer[36], categorised by PACA subtypes, represented as Z scores computed from the distribution of gene expression values across all PACA samples.

of the four classes of cyclins (details in Methods). Our analysis revealed that PACA 1 and 2 tumours exhibit comparable levels of cyclin D1 expression, which is higher than that of the PACA 3 subtype (Supplementary Fig. 6h). Moreover, cyclin D1 expression correlated with mutation burden at origins ($Rho = 0.404$, Supplementary Fig. 6i) indicating that cell proliferation is associated with mutational burden at origins. Additionally, we found that the PACA 1 subtype is characterised by a longer S-phase and shorter G1 and G2 phases compared

to PACA 2 and 3 tumours (Supplementary Fig. 6j). Collectively, these observations demonstrate that increased cell proliferation and cell cycle dysregulation exacerbate the mutational burden at origins.

Our GSEA analysis also uncovered an association between the dysregulation of carbohydrate metabolism pathways and mutagenesis at origins ($P = 9.84 \times 10^{-14}$, permutation test, Supplementary Fig. 6e). To delve deeper into this observation, we examined the specific impact on glycolysis and the pentose phosphate pathway (PPP) in the PACA 1

tumour subtype (Supplementary Fig. 6f). Both pathways are intricately linked and exert significant influence on regulating DNA synthesis and repair in cancer[37]. Glucose-6-phosphate serves as a common substrate for both pathways, contributing to the production of pyruvate and ATP in glycolysis or ribonucleotides in the PPP. Inhibiting ribonucleotide synthesis is expected to affect DNA synthesis and repair[38]. Consequently, we hypothesised that the stimulation of glycolysis and inhibition of the PPP might elevate mutation rates at origins. Indeed, we observed a strong association (Supplementary Fig. 6f), with notable correlations between the expression levels of key enzymes and the ratio of mutation at origins versus flanks (Supplementary Fig. 6g). For instance, both the expression of ATIC (a bifunctional enzyme involved in the final two steps of purine biosynthesis) and TALDO1 (an enzyme responsible for generating ribose-5-phosphate) exhibited a negative correlation with mutational burden at origins ($Rho = -0.403$ and $-0.372$ respectively). The limited availability of comprehensive matched mutation and transcriptomic data from other cancers currently limits our analysis to PACA. While it will be important to extend these observations to cancer types beyond PACA, our findings nonetheless suggest an unexpected interplay between metabolic pathways and mutagenesis at origins in cancers.

## Constitutive origins are hotspots for genome rearrangements

Our previous GSEA findings (Fig. 5d) provide support for the downregulation of pathways associated with DNA replication and repair as a primary factor influencing the mutation rate at origins in pancreatic cancers ($P = 3.73 \times 10^{-9}$, permutation test, Supplementary Fig. 6d). To further explore this result, we dissected these pathways into smaller components and observed that the PACA 1 tumour subtype exhibits downregulation of cell cycle dependent DNA damage checkpoints and downregulation of factors involved in processing DNA single-strand breaks (Fig. 6a). Consequently, breaks originating at origins due to polymerase stalling at G4s are more likely to progress into S phase or mitosis without undergoing repair, a scenario conducive to the emergence of genome rearrangements during cancer progression[39]. Indeed, early replicating regions of the genome[8] and ectopic sites of replication initiation induced by oncogene overexpression[4] have been previously shown to be hotspots for genome rearrangements. We therefore evaluated the prevalence of genome rearrangements by focusing on structural and copy number variants, referred to as SVs and CNVs respectively. Our analysis revealed that PACA 1 and 2 subtypes are characterised by an increase in break ends at origins (Fig. 6b) and a higher number of copy number segments (Fig. 6c) compared to the PACA 3 subtype. These findings suggest that origins may serve as hotspots for genome rearrangement in pancreatic cancers.

Due to the limited number of SVs observed at origins in PACA cancers, we expanded our investigation to characterise structural variation at origins across various cancer types. Our analysis confirmed an increase in break ends at origins pan-cancer (Supplementary Fig. 7a) and we proceeded with establishing the signatures of structural variation at origins and origin flanks using a standardised classification procedure[40]. This approach revealed an enrichment of short tandem duplications at origins compared to their flanks (Fig. 6d). These enriched duplication events occur to the detriment of other SVs (Supplementary Fig. 7b), exhibit a median length of ~100 base pairs (Supplementary Fig. 7c) and stem from break ends occurring predominantly at Gs (Supplementary Fig. 7d). This latter observation underscores the potential role of breaks originating from polymerase stalling at G4s as the primary driver of these events. To validate this hypothesis, we examined the distribution of SV break ends in the vicinity of G4s (Fig. 6e) and found an enrichment of break ends within 5 bp of the start of G4 motifs at origins but not at G4s within origin flanks. Taken together, these findings indicate that breaks arising from replication initiation are transformed into a diverse array of short

tandem duplications, with the potential to alter genome structure and function.

Finally, we characterised the copy number segments mapped within PACA subtype genomes. Notably, we observed that the increase in segment numbers in PACA 1 and 2 (Fig. 6c) coincides with the presence of shorter segments with median lengths of ~2 Mb (Supplementary Fig. 7e), suggesting a potential association between origins and focal copy number alterations. Analysing the pattern of copy number alterations in PACA subtypes, focusing on length, absolute copy number, and potential loss of heterozygosity (LOH) following an established procedure[41], yielded distinct signatures for PACA 1 and 2 when compared to PACA 3 (Fig. 6f). Firstly, PACA 1 and 2 subtypes are characterised by the presence of high copy number segments (3 and above) that underwent LOH, indicative of chromosomal duplication and segmental aneuploidy[41], possibly resulting directly from the replicative stress experienced by these subtypes. More interestingly, these subtypes also exhibited an increase in short segments (1 to 3 Mb) amplified to 3 – 5 copy numbers without signs of LOH. To explore a potential role for constitutive origins in generating these segments, we mapped the distribution of origins within PACA 1 and 2 amplified segments with copy numbers higher or equal to 3 and found that origins marked their boundaries (Fig. 6g). A permutation test confirmed the statistically significant enrichment of origins at the boundaries of these segments ($P \leq 0.001$ over 1000 permutations, Supplementary Fig. 7f). Importantly, such enrichment was not observed for amplified segments from the PACA 3 subtypes (Fig. 6g) or for loss segments across any PACA subtypes (Supplementary Fig. 7g). To better understand the mechanism by which breaks induced at origins drive copy number alteration, we noted that origin enrichment at segment boundaries was confined to amplified segments of 3 or 4 copy numbers (Supplementary Fig. 7h) showing no signs of LOH (Supplementary Fig. 7i). Collectively, these observations support and extend the notion that replication origins are hotspots for copy number alteration and suggest that breaks originating from replication initiation drive focal rearrangements through a specific mechanism discussed below.

## Discussion

Our analysis identifies and characterises biological processes associated with replication initiation that focus mutagenesis and genome rearrangements at thousands of constitutive origins. Notably, while the origin domains we examined cover <0.3% of the human genome, they can contribute up to 2.7% of the mutations in some cluster 2 cancer samples. Considering that constitutive origins are enriched within functional regions of the human genome[7], such as promoters and exon/intron junctions, we anticipate that this focused mutagenesis contributes to the evolution and expression of cancer genomes. Furthermore, we found that origin activity leads to the emergence of numerous tandem duplication events and copy number alterations with potential functional consequences. For instance, we identified the duplication of short origin sequences within the first introns of the MYC and TYK2 proto-oncogenes. These events have the potential to alter the expression of these genes, as origin sequences are enriched for specific transcription factor binding sites and cis-regulatory elements[7,42]. We also mapped the amplification of megabase segments, marked with constitutive origins at their boundaries, containing pancreatic cancer associated oncogenes such as EIF3B or MTA1 that may drive specific cancer phenotypes[43,44].

While the mechanisms underlying some of the mutational processes, we identified are clear, further investigations are necessary to understand the molecular basis for origin-associated genome rearrangements. Although we observed distinctive signatures in SVs and CNVs at origins, the mechanistic basis of these events remains speculative. We have previously demonstrated that breaks are enriched at constitutive origins[7]. Our data supported that such breaks were

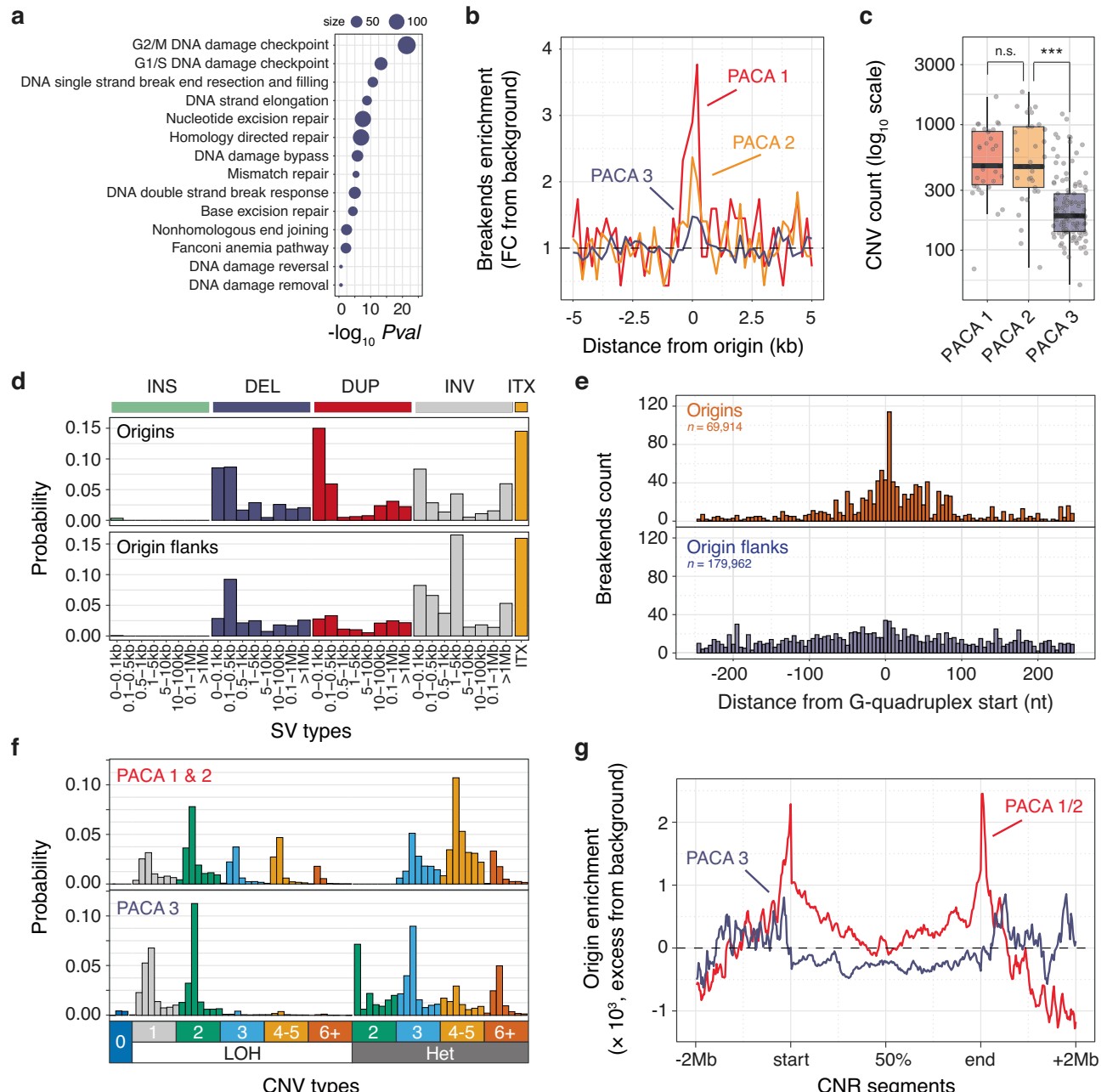

**Fig. 6 | Constitutive origins serve as focal points for structural and copy number variants. a** Analysis of DNA repair and replication pathways. Primary KEGG pathways were dissected into smaller components to identify specific processes downregulated in PACA subtype 1 compared to subtypes 2 and 3. **b** Distribution of structural variant (SV) break ends at origins across PACA tumour subtypes. Enrichment of break ends is presented as the fold change from background calculated from origin flank values. **c** Distribution of the number of copy number variation (CNV) segments in individual PACA samples categorised by PACA tumour subtypes. Box plots illustrate medians and interquartile ranges, with grey dots indicating individual cancer samples. *n. s.* non-significant, ***$P < 0.001$, Kolmogorov-Smirnov test. **d** Pan-cancer signatures of structural variation based on break ends mapped at origins or origin flanks, categorised by SV types and lengths. SV types are: INS (insertion), DEL (deletion), DUP (duplication), INV (inversion) and ITX (translocation). **e** Distribution of SV break ends at 5 nt resolution at

G-quadruplex forming sequences within origin or origin flank domains. **f** Signatures of copy number variation for PACA tumour subtypes, classified by CNV types, copy number, and evidence of loss of heterozygosity (LOH) or not (Het). Each CNV type is further classified by increasing segment sizes (from left to right). Segment sizes were excluded from this plot to enhance clarity. Detailed information on segment sizes can be found in the Methods section. **g** Enrichment of origins within amplified CNV segments in PACA subtypes. Enrichment values were evaluated by partitioning CNV segments and their flanking domains (±2 Mb) into an equal number of windows, calculating the number of origins per segment and per window, and then determining the mean number of origins per window. This value was then adjusted by subtracting the mean values observed in segment flanking domains. The resulting origin enrichment value indicates the excess of origins within a given window.

associated with enrichment of factors associated with non-homologous end joining, which may be expected to favour the observed deletions more than sequence duplications and amplification. Thus, such breaks would not likely explain the association of

constitutive origins and CNVs. However, double-stranded DNA breaks with protruding ends, such as those that may arise following polymerase stalling at G4s near origins, are known to trigger small tandem duplications following overhang gap filling and end-joining (reviewed

in ref. 45). Indeed, secondary structures, particularly G4s, have been associated with CNV boundaries in both cancer[24] and the germline[46]. In this context, the size of tandem duplication events would be controlled by the location of the breaks and the site where DNA synthesis is initiated. Intriguingly, tandem duplications are confined within the 1-kilobase domains defined by replication origins, leading to duplication of origin sequences. Due to the absence of LOH in the resulting amplified segments, we can only speculate that they may result from a mechanism such as fork stalling and template switching (FoSTeS)[47]. In this mechanism, the decoupling between leading and lagging strand synthesis, caused by polymerase stalling at G4s, would trigger a switch to a region with microhomology, such as another replication origin or G4-rich domains, to restart DNA synthesis. Such events could occur multiple times before the replication fork returns to its original position to continue replication to the chromosome end, favouring duplications over deletions and resulting in the amplification without LOH observed in our analysis.

Our findings are based on the comprehensive analysis of mutagenesis occurring at 'constitutive' origins that are conserved across several human cell types[6]. These origins are recurrent sites for replication initiation and accumulate mutations at each cell division. However, it is important to acknowledge the potential variability in the activity and efficiency of these origins across different cancer types. Indeed, certain cancers may inactivate some origins or exhibit variations in their efficiency compared to cultured cells. Additionally, the induction of oncogenes can activate dormant origins, further complicating the estimation of global mutation rates at these sites[4]. Consequently, our estimations may be subject to some degree to both underestimation and overestimation. Nevertheless, our study reveals compelling associations across pan-cancer and tissue-specific analyses, strongly advocating for replication initiation playing a critical role in shaping and focussing specific mutational processes at origin sites. Notably, we demonstrate that the mutagenic potential of replication origins is exacerbated by replicative stress associated with tumour progression. Furthermore, we show that the intrinsic biology of cancer subtypes significantly influences the intensity of mutagenesis observed at these sites. These collective observations underscore the integral role of replication origins as fundamental genetic elements regulating the mutagenesis that drives the development and progression of human cancers.

## Methods

### Cancer mutation and associated datasets

Somatic mutation data for cancer, including single nucleotide variants (SNVs, generally referred to as mutations in the manuscript), structural variants (SVs), and copy number variants (CNVs), were sourced from the International Cancer Genome Consortium (ICGC, release 28). SNVs and SVs for individual cancer samples were derived from the Broad variant calling pipeline, while CNVs for pancreatic ductal adenocarcinoma samples were obtained from the DKFZ/EMBL variant calling pipeline. These mutations were directly used for analysis. Only cancer samples that were whole-genome sequenced and had >5000 mutations were considered for computation of SNV mutation rates, or 50 mutations at origin domains for single base substitution mutation signature analysis. The minor allele frequency was calculated in bins of 100 nucleotides as the number of occurrences of a given mutation divided by the total number of screened cancer samples. Transcriptomic profiles were generated from sequencing-based gene expression data of ICGC tumours, focusing only on protein-coding genes. Gene expression levels were quantified as transcripts per million (TPM) using raw read counts and exon lengths associated with all transcripts mapped to a given gene. Exon lengths were retrieved from the TxDb.Hsapiens.UCSC.hg19.knownGene annotation R/Bioconductor package. Throughout the manuscript we refer to cancer types using the ICGC cancer project nomenclature and their associated

acronyms; BOCA: Bone cancer, BRCA: Breast cancer, BTCA: Biliary tract cancer, CLLE: Chronic lymphocytic leukaemia, CMDI: Chronic myeloid disorders, EOPC: Early onset prostate cancer, ESAD: Oesophageal adenocarcinoma, GACA: Gastric cancer, LAML: Leukaemia, LICA: Liver cancer, LINC: Liver cancer, LIRI: Liver cancer, MALY: B cell malignant lymphoma, MELA: Skin cancer, ORCA: Oral cancer, OV: Ovarian cancer, PACA: Pancreatic cancer, PAEN: Pancreatic endocrine neoplasms, PBCA: Paediatric brain cancer, PRAD: Prostate Adenocarcinoma and RECA: Renal cell cancer.

### Somatic mutation datasets

Somatic mutation data from non-cancerous tissues were obtained from Abascal et al.[13] and Moore et al.[14] Abascal et al. utilised nanorate sequencing (NanoSeq), a duplex sequencing technique with extremely low error rates, to quantify somatic mutations in non-dividing cells across various tissues, including cord blood granulocytes, smooth cardiac muscle, post-mitotic neurons, colonic crypts, and sperm cells. To assess mutational burden at origins, we combined all identified mutations and aligned the signal with constitutive origins. Moore et al. provided data on somatic mutations from multiple cell types in three individuals. These somatic mutations were identified by comparing mutations in 29 anatomical structures from 9 organs to germline mutations, and these data were directly used for analysis.

### Constitutive origin dataset

The constitutive origins analysed in our study were identified using our high-resolution origin mapping technique, initiation site sequencing 2 (ini-seq 2), in the EJ30 bladder carcinoma cell line[6]. These origin coordinates, available from the Gene Expression Omnibus archive with the accession number GSE186675, initially mapped in the hg38 human genome assembly, were converted to the hg19 assembly using the *liftOver* tool from the rtracklayer R/Bioconductor package[48], with chain files obtained from http://hgdownload.cse.ucsc.edu. Previous research has shown significant overlap between these origins and those identified by short nascent strand isolation coupled with next-generation sequencing (SNS-seq) in the same cell line, as well as with a database of 'core' origins identified by SNS-seq in 19 other human cell samples[12]. To ensure accurate comparison of mutation rates, we considered 10 kb flanking regions on either side of origin midpoints. Additionally, to avoid potential biases from replication origin clusters, we focused only on domains containing a single origin resulting in analysis of 9,341 constitutive origins, refer to as 'isolated' origins. The coordinates of origins considered in this work is reported in Supplementary Data 1. To evaluate similarities in terms of SNV count between 'isolated' origins and clustered origins, we define origin clusters as genomic intervals >20 kb that contain at least two constitutive origins. On average, origin clusters contain ~7 origins. To compare mutation rates at constitutive and core origins, we define 'core' origins, in this article, the cell type-independent origins identified by SNS-seq[12] that do not overlap with those mapped using ini-seq 2. We consider two initiation sites to represent the same origin if the mapped domains are located within <5 kb of each other.

### Defining genomic regions and mutation densities

In our analysis, we contrast mutation rates and mutational mechanisms at origin domains with those at their flanking domains. We define origin domains as 1 kb regions centred on origin midpoints, and origin flanking domains as 20 kb regions also centred on origin midpoints but excluding the previously defined origin domains. We calculated mutation rates near origins using 100 bp windows, counting the number of relevant mutations and adjusting for the number of covered bases. To assess mutational burden at origins, we computed ratios of the length-normalised mutation counts at origin domains relative to those in flanking regions.

## Single base substitution mutational signature analysis

Evaluation and visualisation of substitution mutational patterns at constitutive origins were performed using the MutationalPatterns R/Bioconductor package[49]. We defined mutation count matrices by considering the frequencies of each of the six pyrimidine substitutions, calculated in each of the possible 96 trinucleotide 5′ to 3′ contexts. To compute background-adjusted signatures at origins, we adjusted mutation count matrices obtained from origin domains based on their trinucleotide composition and to similar mutation count matrices obtained from origin flanking domains. Tumour sample clustering was performed using these adjusted signatures to differentiate samples according to mutation processes and cosine correlation similarities were used to quantify the closeness of origin-associated signatures. Genome-wide signatures were not corrected for local variation in trinucleotide composition. Origin-associated mutational signatures were compared to known signatures of somatic mutations in cancerous human tissues collected by the Catalogue of Somatic Mutations in Cancer (COSMIC, release v3.2)[15] and available from https://cancer.sanger.ac.uk/signatures/sbs. Exposures to both newly reported and COSMIC signatures were computed in 100 bp windows using the fit_to_signatures and fit_to_signatures_strict function of the MutationalPatterns package respectively. For COSMIC signature refitting experiments, only signatures contributing to at least 50 mutations within at least a window were considered.

## Nucleotide excision repair profiles at constitutive origins

Genome-wide NER sequencing (XR-seq) datasets of cyclobutane pyrimidine dimers (CPD) and pyrimidine-pyrimidone (6–4) photoproducts (6–4 PP) from ultraviolet-irradiated CSB/ERCC6 mutant NHF1 skin fibroblasts[18], were retrieved from Gene Expression Omnibus (GSE67941). To quantify repair against DNase I hypersensitivity (DHS) at origins, ENCODE consortium data for the GM04504 skin fibroblast cell line was used. Sequencing data were formatted into bigwig files to report read counts at 50 nt resolution. These files were used to compute coverage matrices and generate heatmaps using the EnrichedHeatmap R/Bioconductor package[50]. Correlations between mutation counts for cluster 1 tumours, XR-seq, and DHS signals at origins were computed based on averaged signals over origin domains (origin midpoints ± 500 bp). NER strand bias around origins was evaluated by analysing strand-resolved XR-seq signals in 100 bp windows, corrected for dinucleotide base composition. As reported, the MC-062 and 64M-2 antibodies were used for CPD and 6–4 PP XR-seq data generation, respectively, with MC-062 targeting thymidine dimers and 64M-2 targeting UV-C irradiated DNA, in which 6–4 PP adducts are generated at CC, CT, and TC dinucleotides[51]. Thus CPD and 6–4 PP XR-seq data were adjusted for the frequencies of these dinucleotides. To confirm that locally reduced NER activity persists at the origins when controlling for transcription factor binding, we used publicly available ChIP-seq datasets for CTCF, ETS1, REST, EZH2, and NANOG from ENCODE (http://hgdownload.soe.ucsc.edu/goldenPath/hg19/encRegTfbsClustered/) in the GM23338 cell line.

## G-quadruplex forming propensity

To assess the likelihood of a sequence folding into a G-quadruplex (G4) structure and to quantify the predicted stability of the resulting motif, we employed the G4H score, which is derived from the *G4 Hunter* algorithm[25]. This algorithm takes into account the richness and skewness of guanine (G) content within a sequence to compute its G4H score. Each position within a sequence receives a score ranging from −4 to 4. G-skewness is considered by assigning a neutral score (0) to adenine (A) and thymine (T), a positive score to guanine (G) (as it promotes G4 formation), and a negative score to cytosine (C) (as it promotes hairpin and impedes G4 formation). G-richness is accounted for by assigning a score equivalent to the length of G tracts for each G, with a maximum score of 4. Similarly, cytosines in C tracts receive

negative scores. The G4H score for a sequence is the average of these individual scores. Thus, the propensity of a sequence to fold into a G4 motif correlates with its G4H score, with higher scores indicating G4 formation on the plus strand and lower scores on the minus strand. To investigate G4 formation at specific positions, we extracted sequence contexts (± 25 bp) and computed associated G4H scores. To identify potential G4-forming sequences, we searched for sequences conforming to the regular expression pattern $N_5G_{3+}N_{1-12}G_{3+}N_{1-12}G_{3+}N_{1-12}G_{3+}N_5$ (where N represents any base), which represents a slight modification from the known *quadparser* algorithm[26], allowing for extended loop lengths. Additionally, we expanded the core motif for G4 formation by including five flanking bases to account for the G4 context when predicting stability using the G4 Hunter algorithm. Experimentally validated G4 structures within cell chromatin were identified from G4access peaks. G4access is an antibody- and crosslinking-independent method used to isolate and sequence G4s associated with open chromatin through nuclease digestion[27]. In our analysis, following the approach of the original publication, we considered G4-forming sequences as the 25 nt sequences with the highest G4H score within G4access peaks from HaCaT, K562, and Raji cells.

## Differential gene expression and gene set enrichment analysis

Differential gene expression analysis was performed using the DESeq2 R/Bioconductor package[52], with pancreatic ductal adenocarcinoma (PACA) samples grouped based on cancer subtypes identified through UMAP dimensionality reduction of their transcriptomic profiles. UMAP analysis was performed using the umap R package[53] from TPM values of protein coding gene expression using two components and starting from 20 random states. Changes in gene expression were computed by contrasting PACA subtype 1 with PACA subtypes 2 and 3 independently. Differentially expressed genes were ranked using the DESeq Wald statistic and subjected to gene set enrichment analyses (GSEA) using the fgsea R/Bioconductor package[54] and curated KEGG pathways. Hallmark gene sets sourced from the Molecular Signatures Database (MSigDB, version 7.1)[55] were used, available as R lists in RDS format at https://bioinf.wehi.edu.au/MSigDB/. Pathway enrichments were determined by comparing PACA subtype 1 with PACA subtypes 2 and 3 individually, followed by averaging normalised enrichment scores and combining P-values using Fisher's method. Pathways of interest were further dissected into smaller components using information from the Reactome pathway database (https://reactome.org/)[56]. Once more, normalised enrichment scores and associated P-values were combined from independent analyses comparing PACA subtype 1 with PACA subtypes 2 and 3. Consequently, our analysis delineates pathways that are specifically dysregulated in PACA subtype 1, which exhibits a higher mutational burden at origins.

## Replicative stress and cell cycle evaluation in pancreatic ductal adenocarcinoma

Replicative stress within PACA samples was evaluated from the gene expression profiles of established biomarkers[36]. Expression levels of these biomarkers were computed as Z scores relative to the distribution of raw count values across all PACA samples. Analysis of cell proliferation and cell cycle dysregulation in PACA samples was inferred from the expression patterns of the four cyclin classes: cyclin A (CCNA1 and CCNA2), cyclin B (CCNB1 and CCNB2), cyclin D (CCND1, CCND2, and CCND3), and cyclin E (CCNE1 and CCNE2) in transcripts per million (TPM). These expression profiles were used to estimate the cell population proportions in G1/S phase (calculated as the ratio of cyclin E expression to the sum expression of all cyclins), S/G2 phase (calculated as the ratio of cyclin A expression to the sum expression of all cyclins), and G2/M phase (calculated as the ratio of cyclin B expression to the sum expression of all cyclins)[57].

## Signature of structural variation

Pan-cancer structural variant (SV) breakpoints were mapped to either origin or origin-flanking domains, and SV signatures within these domains were extracted following a standardised protocol[40]. SVs were annotated as INS (insertion), DEL (deletion), DUP (duplication), INV (inversion), and ITX (translocation) based on the orientation and position of the genomic location covered by the supporting reads, using a custom R script. Additionally, SVs were further classified based on their length into eight predefined intervals: 0–0.1 kb, 0.1–0.5 kb, 0.5–1 kb, 1–5 kb, 5-10 kb, 10–100 kb, 0.1–1 Mb and >1 Mb. Finally, the frequency of occurrence for each SV event was computed.

## Signature of copy number alteration in pancreatic ductal adenocarcinoma and relationship with constitutive origins

To identify signatures of copy number alteration in PACA subtypes, we adapted an established procedure[40] with slight modifications to emphasise focal amplification of small segments. We combined segments from PACA subtypes 1 and 2 as they did not show significant differences in a preliminary analysis. Annotations from the ICGC data describing mutation types were used to evaluate loss of heterozygosity (LOH) or heterozygosity (Het) within CNV segments. Subsequently, segments were categorised based on their absolute copy number and length into seven predefined intervals: 0–0.1 Mb, 0.1–1 Mb, 1–2 Mb, 2–3 Mb, 3–5 Mb, 5–10 Mb, and >10 Mb. The frequency of occurrence for each copy number variation (CNV) event was determined. We investigated the association between constitutive origins and copy number alterations by examining the distributions of origins within copy number segments. We first computed the number of origins per 100 kb window across the human genome, generating a bigwig file for subsequent coverage calculation at CNV segments. We then evaluated origin distribution by dividing CNV segments and their adjacent domains (±2 Mb) into 100 equally sized windows and determining origin coverage per segment and per window using the normalizeToMatrix function of the EnrichedHeatmap R/Bioconductor package[50]. Enrichment values were then computed by averaging origin counts across windows within segments of interest. These values were adjusted for background by subtracting the mean counts observed in flanking domain segments. Consequently, the reported enrichment values represent the excess of origins compared to background levels. CNV segments with length >100 kb and <5 Mb were considered for these analyses. To assess origin enrichment at the boundaries of amplified segments in PACA 1 and 2 cancer subtypes, we identified segment breakpoints as the start or end of mapped amplified segments. We evaluated the significance of overlap with origin domains using the *overlapPermTest* function of the regioneR R/Bioconductor package[58], conducting 1,000 permutations.

## Computational and statistical analyses

Analysis and all statistical calculations were performed in R (version 4.0.3). R packages: MutationalPatterns (3.15.0), DESeq2 (1.45.0), StructuralVariantAnnotation (1.22.0), dendextend (1.17.1), umap (0.2.10.0), EnrichedHeatmap (1.35.0), regioneR (1.38.0), rtracklayer (1.65.0), biomaRt (2.61.0), BSgenome.Hsapiens.UCSC.hg19 (1.4.3), org.Hs.eg.db (3.19.1), TxDb.Hsapiens.UCSC.hg19.knownGene (3.2.2), tidyverse (2.0.0), tidyr (1.3.1), dplyr (1.1.4), tibble (3.2.1), stringr (1.5.1), readxl (1.4.3), rlist (0.4.6.2), ggplot2 (3.5.1), ggpubr (0.6.0), scales (1.3.0), plotrix (3.8-4), forcats (1.0.0), gplots (3.1.3.1), wesanderson (0.3.7), RColorBrewer (1.1-3).

## Reporting summary

Further information on research design is available in the Nature Portfolio Reporting Summary linked to this article.

## Data availability

The data analysed in this manuscript was obtained from the ICGC Data Portal. This portal was officially closed in June 2024. However, the latest data release is accessible via an SFTP server. Access instructions can be found in the Legacy ICGC 25 K Data documentation at https://docs.icgc-argo.org/docs/data-access/icgc-25k-data#accessing-icgc-25k-release-data. The Ini-seq2 replication origin BED files[6] can be found at NCBI's Gene Expression Omnibus (www.ncbi.nlm.nih.gov/geo/) under GEO Series accession number GSE186675.

## Code availability

The Bash and R codes used to analyse the data and generate the figures reported in the manuscript are available at https://github.com/Sale-lab/OriCan[59].

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

## Acknowledgements

We thank I. Clayson, T. Darling and J. Grimmet in Scientific Computing at the LMB for support. Work in the Sale group is supported by a core grant to the LMB by the MRC (U105178808).

## Author contributions

The project was conceived by P.M. and J.E.S. P.M. was responsible for conceptualisation, methodology and formal analysis. P.M., G.G. and J.E.S. contributed to data interpretation. P.M. was responsible for writing the original draft. P.M., G.G. and J.E.S. reviewed and edited the manuscript.

## Competing interests

The authors declare no competing interests.
