## [Transparent Peer Review file · Nature Communications]

DNA replication initiation drives focal mutagenesis and rearrangements in human cancers

Corresponding Author: Dr Julian Sale

Parts of this Peer Review File have been redacted as indicated to maintain confidentiality.

Version 0:

Reviewer comments:

Reviewer #1

(Remarks to the Author)

In this study, Murat et al. comprehensively address the occurrence of various mutational processes locally at human origins of replication. Work addressing this type of topic is timely, since the mechanisms generating the mutation rate heterogeneity across the human genome are understudied. A recent publication by the same authors addresses effects of origins on local mutation rates in the germline genome, and here they turn to somatic genomes which enable conveniently studying a wider diversity of mutagenic processes, less confounded by negative selection. They identify that some of the known processes are enriched at origins (UV, AID) and they identify some probably new processes (SBS43 mutations, certain rearrangements – short duplications) which occur apparently exclusively in hotspots. Mechanistic insight is gained via analysis of repair maps, local DNA motifs (AID context, G4) and gene expression analysis for inference of subtypes and cell proliferation. Overall the study has lots of interesting elements, however the various findings reported do need additional support, in our opinion, in form of analyses to rule out various confounders of local mutation rates, replication in additional cancer types, more careful association analyses to gene expression, and considering to include analysis of various other experimental data sets. Please see the remarks below.

Major:

1. A general concern about the study is that there is not clarity to what extent the DNA replication initiation itself, rather than some other process occurring at the same locus, is causal to the changes in mutation rates. There could be better controls for known confounders. It would be important to provide better evidence for this, since causal language is used in the text. In particular:

1a. regarding the NER deficiency and skin mutations. Can we see evidence that locally lowered NER remains seen at the origins, even when controlling for binding of CTCF and of transcription factors? (in line 150 they actually mention „loading of the pre-replication complex“ as causal, but there was not a test with pre-replication complex binding sites) Does UV mutagenesis at origins remain high after removing the ETS sites?

1b. regarding the widespread mutagenesis in Cluster 2 containing various cancer types. G4s are proposed as causal to this mutagenesis. One way to substantiate this is to monitor if mutation rates at origins become unremarkable upon excluding the G4 regions. (see also comment 3b below). Alternatively or in addition, controlling for other possible confounding factors at origins would help.

2. The set of origins studied (iniseq2 from a bladder cell line) may or may not be well representative; the different techniques to identify origins have different biases, and as they mention in the discussion there would also be some differences between tissues and between healthy and transformed cells.

2a. While this is addressed textually in the discussion, it would be better addressed with actual data. Does using e.g. consensus SNS-seq origins (or another dataset they find appropriate) give similar results?

2b. Also, regarding excluding clustered origins; we understand the rationale for doing that but have a slight concern this could introduce a bias, since the clustered origins were said to have distinct biological properties.

3. Regarding using pancreatic cancer to study the association between origin-focused mutagenesis, and gene expression programs. A good justification of why only this cancer type is analyzed is lacking. Even though PACA is one of the cancer types with high variability in mutagenesis and a high number of samples, there are other cancer types such as BRCA with similar characteristics and for which it is known that there is a subtype division.

3a. Would it be possible to perform this subtype analysis in one or more other cancer types with enough sample size and report their enrichment and GSEA in supplementary data? In particular Whatever associations they found in PACA e.g. with Cyclin D gene expression, or certain pathways, would need to be shown in at least 1 other cancer type, or in a pan-cancer analysis, to be plausible. It seems unlikely that the SBS43-like origin mutagenesis that they identified in Cluster 2 (which contains many tissues) would be confined only to pancreatic cells.

3b. The more fine-grained analyses of gene expression (meaning: beyond looking at the broad clusters, and focussing on specific pathways or individual genes), like those described on line 335, have potential to be confounded by sample purity, and by aneuploidies. If they want to draw firmer conclusions from this expression correlations (e.g. for glycolysis and pentose phosphate pathways lines 310-326; or downregulation of DNA ss break processing factors lines 335-340), controls for these confounding factors would be warranted. Else these claims would be better marked as speculative, and relegated to supplementary text or discussion or dropped altogether.

3c. In addition, it is known that many mutational signatures are tissue-specific to some extent. In the clustering step, would they get the same signatures if clustering the samples according to the 5 clusters but also by cancer type? Finding the same signature (i.e. cluster 2 signature) in several cancer types will show that it is robust.

By eye it seems that there are subclusters within cluster 5. Have the authors tried to split it?

4. To support mechanisms, DNA sequence analysis is used as a proxy for comparing to experimental data, even when the experimental data is available. Two examples of this:

4a. Regarding AID, the tetranucleotide enrichment is fine, but much stronger evidence that oris are AID off-targets would be simply to check overlap with some of the available experimental datasets.

4b. Regarding G-quadruplexes: they can indeed be predicted from sequence using a pattern-matching approach, however they were measured experimentally as well (wherein some measurements point to examples that would not be seen with the pattern matching approach).

5. Regarding the methodology, the authors background-adjust the signatures to spot mutational signatures at the origins location specifically. We understand that they need to do this because they extracted the signatures by clustering the profiles directly. However, if a factorization method (such as NMF) is used to extract signatures, the algorithm should directly separate if more than one process is affecting the mutational landscape at origins into different components, possibly reducing the need to background-adjust the profiles according to the flanking regions. Could the authors clarify/explain why they used this clustering approach rather than a more standard approach as it will be applying NMF and state the advantages and caveats in the text.

Minor:

- Linking the origin-specific signatures activity to external/internal mutagenic exposures may be informative of mechanisms of some of the enigmatic processes occurring at origins. This is related with line 86. „These observations suggest that increased mutation rates at origins are driven by mutational processes of varying degrees of intensity rather than inherent tumour properties“ and line 119 „We then compared these origin-associated signatures with genome-wide signatures derived from samples within the respective clusters (Extended Data Fig. 2b).“ Is there an analysis that compares the activities („exposures“) of known SBS signatures genome-wide with the origin-specific signatures, across tumor samples in each cluster? Ext Fig 2b is related but I think not quite that.

- Lines 75-76 „we corrected the apparent mutation rates of each of the six pyrimidines mutation by their occurrences and found increased mutation rates associated to each type (Fig. 1a).“ I presume this is control by trinucleotides? If not, it should be performed; if it is, then make it clear in the text.

- Lines 112-116 „Cluster 2 signature exhibits high similarity to SBS43 (cosine similarity = 0.95), a signature of unknown aetiology,“ COSMIC states SBS43 may be an artefact – any comments on this? „Cluster 3 signature shares moderate similarities with the known SBS10b signature (cosine similarity = 0.64, associated with polymerase epsilon exonuclease domain mutations).“ 0.64 is very low similarity and so it is not likely to link the mechanism to SBS10, so writing this could be misleading.

- Line 357 „Taken together, these findings indicate that breaks arising from replication initiation are transformed into a diverse array of short tandem duplications,“ did they consider that these duplications might result from re-replication (where a break, in one interpretation, would not be considered an initiating event). Could merit discussing.

- Line 418 in Discussion. Could they clarify how does this hypothetical mechanism of switching to another region w/microhomology cause duplications but not deletions?

- Line 435 in Discussion „intrinsic biology of cancer subtypes significantly influences the 435 intensity of mutagenesis observed at these sites.“ is overstating the impact of the data, if this remains shown only for one cancer type as currently (pancreatic).

- How the “Minor allele calculations” are performed seems not explained in the methods

- Line 81 to 84 when they compare the mutation rates at origins they say that cancer types such as gastric and gallbladder have high mut rates based on fig 1b, but when they normalized the mutation rates by the flankings (results in fig 1c) we can see that the gastric cancer GACA (stomach) is below 1 meaning its relative mutation rates its not high. It is confusing that it is used as an example of high mutation rates in the text.

(Remarks on code availability)

Reviewer #2

(Remarks to the Author)

In this work, Murat et al. go forward from their 2022 work focusing upon constitutive origins (COs) in humans to try and find mutational processes at replication origins. They use single nucleotide variants from across the ICGC pancancer cohort, deriving ratios of mutations between origins and their flanking regions binning them into characteristic clusters, before attempting to resolve the mechanisms underlying profiles in some clusters. While there's interesting work here, their analyses are flawed, relying upon poor statistical assertions to knit together too many wide-ranging analyses. Fundamentally, their work is undermined by well-understood alternative explanations that are ignored but lie at the bases of their analyses. It is clearly a work of great effort, but I think a significant change in narrative angle and better delivery, especially in use of statistics guiding the narrative, is needed.

Deriving COs was initially to acquire a set of dependable origins with which to generalise with across tissue types but then to break them down in this way doesn't make sense- yes, the CO come from those in the bladder cancer cell line and the cell lines used by Akerman, but it's highly probable that differences in expression and a permissive chromatin environment impact DNA damage and repair associated with various mutational processes that are highly likely to coincide positionally with stable origins. For example, Conrad Nieduszynski just just this as part of a recent preprint where they propose that transcription tightly confines replication at active genes <https://www.biorxiv.org/content/10.1101/2024.04.28.591325v1.full.pdf>. In the 2022 paper, the functionality of included COs is well examined, but is missing here, and while I would expect to be told to refer to that paper, it looks like the set of COs is derived in the same way but is less than half the size.

A big red flag associated with this is that the cancer sampling clusters in Fig 2a are only apparent in very particular cancer types. Cluster 1 is driven by UV damage, obviously, and cluster 4 is something particular to lymphoma (MALY). Though cluster 2 is composed of multiple cancer types, it could be driven by one or two cancer types. Debatably, all of these clusters are driven by a small number of origins. You could do a much better job unpicking this, for example, plotting 1b for samples at specific COs that contribute to specific clusters against their other COs and at COs in other cancers. This would be instead of the way you use mutation spectra, which has created a biased analysis. Presenting the data from 1a, such as providing the CO counts (or even as bed files) and sample counts in a table would be useful to assess this. Importantly, regression would be an easy and powerful way of asserting that the enrichment of mutations at origins is driven by particular cancer types or their properties, such as the data in Extended Data Figure 1d, where total genomic burden is plotted against the ratio of mutations at origins against their flanks and gives a really strange distribution.

This could also improve how you pull apart the data to make assertions, which you occasionally make with quite dubious correlation coefficients. For example, at L90 you demonstrate a strong correlation between the total number of mutations and those at origin, where $R=0.954$, but then go on to say that the origin/flank ratio is the opposite of this, when $R=-0.246$, is not a fair comparison. Your reported correlation coefficients from L152 are not justifiably extreme enough to make the assertions about NER that you make: 0.4, 0.17, -0.44 and -0.36. Relating to this, on L85 and L128 you say differences are significant, but there are no associated statistics. In association with this, I don't like that the KS test is used when a Mann-Whitney is more appropriate. The KS is also less conservative and more likely to give a significant P value.

Another line of evidence propping up the analysis in the early stages is use of mutational spectra, though this is deployed clumsily. For example, in L122, you say that the lack of resemblance to extant signatures validates your approach to looking for mutagenic processes at origins, but the set up is biased in a way that will fail to resemble known signatures. This is demonstrated in EDF 2b, which I actively dislike as, of course multiple signatures packed on top of each other, consisting of multiple mutational processes, will generate weird signatures, even if they're across the same sorts of genomic feature. To do this properly in the future, you should consider de novo signature deconvolution and fitting. It's worth noting that, similar to my previous statements about correlation, on L115 you cite a cosine similarity of 0.64, but this is too low to be considered similar to the "matching" COSMIC signature. Another concern is that, on L112 you consider SBS43 to be relevant, though this is a sequencing artefact that isn't included in COSMIC or the PCAWG papers.

To go back to other mechanisms that potentially explain the mutational patterns these authors are seeing, I'll focus on cluster 1. It's well known that DNA damage from UV is occluded by bound transcription factors and therefore has elevated mutation rates. While you acknowledge this in L149, you don't do anything to demonstrate that the COs responsible aren't at transcription start sites of genes expressed in the relevant cell of origin. In addition, there isn't statistical analysis to assert that NER activity is different between the leading and lagging strands. This finding is odd, given that it was generally accepted that there were lower mutation rates on the leading strand, and more recent evidence has demonstrated that this effect is the result of biases in expression. Nuria Lopez-Bigas' group has performed relevant analyses and has nicely demonstrated how to effectively compare against expected values that would strengthen the analyses here: <https://www.sciencedirect.com/science/article/pii/S009286741930234X>

I don't deny that the other analyses have merit, but given the poor bases from which these are formed, it's not worth going further. I think it would be better to frame this paper as the distinction between aspects of certain COs and proving that elevated mutation rates are not just incidental to other damage and repair processes. It would also be useful to try and expand your analyses into a few cell lines with consistent origins in early replicating regions. For example Petryk et al (<https://www.nature.com/articles/s41586-020-1943-3>) report ok-seq from two reps of B cells and there's plenty of WGS from lymphomas available, all highly related to your MALY story.

I truly hope to see a much improved version of this work in the future.

[redacted for confidentiality]

I've included other notes that I made going through the manuscript, which I hope are useful:

The number of origins under consideration isn't always clear and might be insightful. A proper definition of constitutive origins is needed in the main body of the text.

L68 It would be excellent to include an approximate description of the sites you're looking at initially- how big, how many and how conserved, though this latter point could be included in the introduction.

L75 It's standard for the field, but unclear to the uninitiated that these mutation classes include their complements, e.g. T>A also represents A>T.

L150 This is also explained by TF binding of adducts, occluding repair.

L214 stem, not stems

L256-258 reference needed

L265 Why these tumours? What links them? Pulling apart effects by cancer type will not only get towards a mechanism.

PACA story - 8oxoG would likely be enriched at G4 sites. Can you discount that there is an enrichment at COs compared to other parts of the genome?

Is G4 formation enriched at COs compared to other origins and the rest of the genome?

Figures general statements: I would smooth your line plots to make it easier to see the profile of effects, e.g. 2d and EDF 2C. You've got error bars on lots of plots, but I don't know what these mean.

Fig 1 legend Add fold change (FC)

Stats for 2e and f would be nice, to see if they're significantly enriched.

Fig4b and c, in other plots the points are overlaid on top of box plots, the number of origins in each category would at least be useful

Fig4e How do you get negative contribution from SBS10c?

EDF1c and d What are the error bars?

EDF1d 1c goes up to 5, is the data in EDF1d not the same?

EDF2a these are in a not very helpful order

Fig5d What lies in the extremes of the distribution? A table of them all would be interesting.

Data availability: the link for the code associated with this work doesn't work. I would expect a bed of the processed COs to be available too.

Occasionally, clusters are grouped for convenience, rather than the distinction appreciated. For example, Clusters 1, 3 and 5 are re-plotted together, presumably to emphasise the peak at the origin in cluster 1, which would be diminished when plotted with clusters 2 and 4. Defining PACA 2 and 3 as distinct and then ignoring that in the analyses looks bad.

(Remarks on code availability)

Reviewer #3

(Remarks to the Author)

The authors explored the role of DNA replication in driving mutagenesis and genomic rearrangements across various human cancers. The study comprehensively analyzed mutation signatures at the consecutive replication origins. They analyzed mutation calls from the International Cancer Genome Consortium aggregated from 86 cancer projects. They found that sites of replication initiation act as hotspots for mutagenesis in human cancers. Subsequently, the authors focused on tumors with over 50 mutations at origins (227 tumors across 17 cancer types) and characterized the mutation patterns around the origins. These origins are associated with increased mutation rates compared to its 2.5kb proximal flanking regions and genome-wide. They identified five mutation patterns enriched at the replication origins, of which clusters 1 and 4 are specific to melanoma and malignant lymphoma, respectively. They found that the aggregate origin signatures derived for clusters 3 and 5 show no discernible contribution to origin mutagenesis, leaving mutation cluster 2 (enriched in pancreatic cancer, esophagus adenocarcinoma, and breast cancer) as a newly identified mutation signature associated with replication initiation.

The authors then conducted a series of analyses linking mutation signature clusters to mutation processes. They found that mutation cluster 1 resembles COSMIC SBS7b, a process linked to UV light and nucleotide excision repair. The author proposed that this resemblance is due to impaired NER at the origin. Next, the authors analyzed the XR-seq generated using human fibroblasts NER accessibility at the replication origins. They found that mutation density at origins correlates negatively with NER activity, suggesting that replication origins inhibit NER (although it remains unclear how TT dimers were transformed into C mutations). The authors then investigated the process for mutation cluster 4, finding this cluster significantly similar to SBS84, which was proposed to depend on AID. They concluded that the replication origins are off-targets of AID. The authors also investigated mutation cluster 2, finding that T to G mutations at the origin occur within G-rich sequences, particularly embedded within G tracks.

This paper demonstrates that each cancer type has distinct mutation signatures around the replication origins. With cluster 2 as a newly identified mutation signature at the origins, enriched in three major cancer types, this manuscript offers new insights into cancer genome evolution.

The reviewer noted one major shortcoming in the current manuscript. Given that the authors analyzed mutations around the constitutive origins, the findings that each tumor type displays different mutation signatures are very interesting. The authors have implied that these are tissue/cell-type specific lesions. Considering that new technologies have significantly improved sequencing accuracy (e.g., Nanoseq), the authors could address the mutation signatures at or around replication origins in the precancerous stage or normal tissues to clarify whether these mutation clusters are cancer-specific. These additional analyses and clarifications would significantly enhance the manuscript's contributions to understanding the complex dynamics of DNA replication, repair mechanisms, and mutagenesis in human cancers.

In addition, the author also found that replication origins are hotspots for genome rearrangement. This notion has been mentioned previously by Barlow et al. (<https://doi.org/10.1016/j.cell.2013.01.006>) and Macheret et al. (<https://doi.org/10.1038/nature25507>), where early replicating replication initiation sites are hotspots for DNA breaks and genome rearrangement. The author should further specify how their contribution differs from previous publications.

Other comments:

1. The authors have indicated that nucleotide excision repair (NER) activity is generally suppressed at the origin of replication. It would be beneficial if the authors could speculate on potential reasons for this suppression. Additionally, considering other mechanisms that could contribute to the C to T mutation beyond NER would enrich the discussion. The manuscript also mentions activity at the lagging strands near the origins. Clarification on whether this is a property specific to lagging strands or unique to replication initiation zones would be valuable.

2. Regarding the AID-dependent mutation signature identified in cluster four, it is noted that AID is predominantly expressed in B cells. This raises questions about the presence and implications of such mutations in both B and T cell lymphomas.

Given that approximately 50 off-target sites for AID have been identified in mouse B cells (<https://doi.org/10.1016/j.cell.2011.07.049>), which are hotspots for translocations, a comparison of AID's contribution at the origins to other AID off-target sites in B cell lymphoma, genome-wide, would be insightful. Furthermore, the base excision repair pathway removes deaminated cytidine at AID targeting sites, leading to DNA double-strand breaks and potential translocations. It would be valuable if the authors investigate whether more translocations are observed around the origins compared to the rest AID off-target sites in the genomes.

3. The authors conclude that increased mutation rates at origins may result from replicative stress induced by excessive cell proliferation based on GSEA analysis findings related to the MAPK and JAK-STAT pathways. However, these pathways are not exclusively involved in regulating cell growth, DNA replication speed, or the number of additional origins fired in cells. To strengthen the assertion that replication stress or cell proliferation causes the mutation cluster type at the origin, the authors are encouraged to explore other datasets that directly test mutation density under experimentally controlled conditions altering cell growth/proliferation. Examples of such conditions in normal cells include sonic hedgehog activation in granule neuron progenitors (<https://doi.org/10.1038/s43018-020-0094-7>), and erythroid progenitors (<https://doi.org/10.1126/sciadv.1700298>).

4. Regarding methodology and details in analytic approaches, the authors should, perhaps in supplementary information, provide detailed versions of R and packages. In line 581, please describe "a standardised protocol" to clarify the rationale and specifications for readers who have not done this type of analysis before.

5. Across the manuscript, the "online methods" were mentioned several times while the reviewers did not have access to this method section. Please make sure this information is available.

6. At "Data availability," the link to <https://github.com/Sale-lab/OriCan> was not found.

7. Figure 1, legend: please describe what "FC" stands for.

8. Figure 2e, legend: please explain the Y axis "log10 scale". Is the scale in log10, or does it mean something else?

9. Figure 3, legend: please spell out the full name for "CPF" and "PP" in the legend.

(Remarks on code availability)

The link to the code (<https://github.com/Sale-lab/OriCan>) cannot be found.

Reviewer #4

(Remarks to the Author)

(Remarks on code availability)

Version 1:

Reviewer comments:

Reviewer #1

(Remarks to the Author)

In the revised study, Murat et al. have performed various additional analyses prompted by our remarks and those of other reviewers. Overall, we think the central conclusion of the study is substantially better supported after these revisions:

- some possible confounding factors have been ruled out, most prominently binding of TFs and CTCF
- G4s were further supported as causal to a part of the mutagenesis, and the G4 features leading to higher mutagenicity were elucidated
- an important adjustment for local trinucleotide composition at and around origins was implemented
- the new A>C (T>G) signature in the multi-tissue "cluster 2" was shown to be operative across many individual cancer types
- origins found by SNS-seq outside of the original ini-seq2 detected set were shown to be (modestly) mutagenic as well; both clustered and unclustered origins were mutagenic
- the SBS43-like signature observed has some more support that it is not an artefact, on the basis of tissue-specificity and association with active origins rather than only the -G4 repetitive sequences

There are two outstanding issues from our side. Importantly, these are, in our opinion, not central for the conclusions of the study. Therefore, we would leave it at the authors' discretion to decide if these should be addressed. Even if these issues are not addressed, the study would be sufficiently solid, and findings robust, in the current state. We would be enthusiastic about seeing it published.

1- The subtype analysis, performed originally only on pancreatic cancers (PACA), based on the gene expression patterns. For these findings to be shown to be generally valid across cancer types, additional somatic tissues would need to show similar trends. While it is fine to present this result as-is -- clearly stating it is currently limited to PACA as other cancer types were not studied -- we do draw the authors' attention to the Hartwig Medical Foundation cancer WGS dataset, which has transcriptomes for ~3k samples currently. This would allow better testing of the proliferative driver/replicative stress hypothesis, even if in future work.

2- The mutational signature analysis, which was suggested by both us and by Reviewer 2 to be better addressed with a more standard, factorization (e.g. NMF) based approach, rather than a simpler approach they took by clustering the tumors based on the mutational spectrum clustering. Overall, we think the current approach is sufficient, and perhaps more appropriate with low mutation counts (their genomic regions-of-interest are short, which limits the number of mutations available). We do not think the main conclusion about increased mutagenesis at frequently-used origins would change if NMF were employed. It is possible they would recover a cleaner signature and/or multiple signatures, though; again as above we leave it to authors discretion to implement this change, or to leave it for future work.

~~

Additionally, we provide our thoughts on the comments by Reviewer 2 below, as prompted by the editor. In our opinion, some general concerns of Reviewer 2 were not necessarily justified:

1- the reviewer 2 stated "work is undermined by well-understood alternative explanations that are ignored but lie at the bases of their analyses" but did not state clearly what these alternative explanations were. It is possible they meant the (later mentioned) "differences in expression and a permissive chromatin environment impact DNA damage and repair associated with various mutational processes that are highly likely to coincide positionally with stable origins". Differences in expression (gene-resolution, 1-10kb) cannot explain the narrow peaks in mutagenesis (roughly 100 nt-resolution). Permissive chromatin environment as a factor seems to be ruled out by the new analysis that binding of certain TF does not explain the mutagenesis at origins. This seems to not be a concern.

2- the reviewer 2 stated "their analyses are flawed, relying upon poor statistical assertions" but apparently, to our understanding, did not show where the statistical analyses were flawed. This may have referred to uses of the K-S or Mann-Whitney test, which is in our opinion not a substantial enough difference to dismiss the result.

This may also have referred to "all of these clusters are driven by a small number of origins", a comment addressed adequately by Rebuttal figure 9, which provides transparency on the number of origins. In summary, we do not share these worries about the study.

Some specific concerns of Reviewer 2 were valid critique. In our opinion, they were answered adequately by the authors of the study in their rebuttal. In specific:

3- Usage of one specific set of "constitutive origins" (iniseq2 derived) yet not other sets available in the literature (e.g. Nieduszynski paper). We would agree with the responses of the authors that various methods to identify origins such as the recent Nieduszynski methods, or OK-Seq or Bubble-seq would have either an insufficient power (former) or too low resolution (latter 2 methods) to be useful for their analysis of focal mutagenesis at origins. Therefore, it does not make too much sense to consult them here.

The one origin identification method that is useful for their analysis is SNS-seq; indeed in the revision they do measure mutation enrichment at SNS-seq peaks that were not already in their iniseq2 origin dataset, finding a modest enrichment of mutation rates. This supports their general claim that origins are mutagenic, and additionally supports that the less-used origins (in SNS but not already in iniseq2) are the less-mutagenized ones. Overall, we think the authors did their "due diligence" regarding replication origin data sets considered, and that the iniseq2 set of origins they employ would be sufficient to support their main hypothesis.

4- Regarding analysis of mutation spectra. Reviewer 2 mentioned "the way you use mutation spectra, which has created a biased analysis" and "the lack of resemblance to extant signatures validates your approach to looking for mutagenic processes at origins, but the set up is biased in a way that will fail to resemble known signatures" and "multiple signatures packed on top of each other, consisting of multiple mutational processes, will generate weird signatures, even if they're across the same sorts of genomic feature. To do this properly in the future, you should consider de novo signature deconvolution and fitting."

In this remark, we would agree with the reviewer 2 in principle; we did also ask a related question (question 5 in the original review) as why the more standard NMF or similar factorization method was not applied, as opposed to simply clustering the genomes of the spectra. The authors declined to implement this change. Upon consideration, we think implementing this change (while it would bring the study more in line with standard methodologies) is not crucial for the conclusions of the paper to stand. The clustering approach on mutational spectra, even though less powerful than NMF, does require fewer data points to work well and so might be advantageous for very narrow regions, such as the replication origins they analyze herein.

~ end ~

(Remarks on code availability)

Reviewer #3

(Remarks to the Author)

The reviewer acknowledges that the authors have addressed most the concerns raised. The revised manuscript now includes additional analyses that strengthen the authors' arguments, particularly regarding mutation signatures at replication origins in non-proliferating and pre-cancerous tissues. The reviewer appreciates this effort, as it is crucial to establish that these are indeed cancer-specific events.

The reviewer understands that certain points could not be fully addressed due to the limitations caused by the dependence on existing high-quality datasets. As these challenges are beyond the scope of the manuscript and a single lab, they should not be viewed as a weakness.

The inclusion of new data from publicly available datasets, as well as the discussion on how the authors' work diverges from previous studies, adds substantial value to the manuscript. The manuscript should be accepted for publication.

The reviewer found some minor issues in the Github page, which should be further clarify to be usable and produce reproducible results.

(Remarks on code availability)

The link is now available, but it is missing important information, particularly the R version and package dependencies. The data source is not indicated anywhere. Some links are broken (e.g., https://dcc.icgc.org/releases/release_28). Additionally, the link to the reference genomes is missing. The code also needs cleaning, and paths to local datasets should be renamed to something more meaningful and understandable for users.

We did not perform test run for these codes, my lab is lacking capacity to do so at the moment.

Reviewer #4

(Remarks to the Author)

I co-reviewed this manuscript with one of the reviewers who provided the listed reports. This is part of the Nature

Communications initiative to facilitate training in peer review and to provide appropriate recognition for Early Career Researchers who co-review manuscripts.

(Remarks on code availability)

Below, we provide detailed responses to each point raised by the reviewers and outline the resulting changes made to our manuscript. We believe these revisions have strengthened the conclusions of our work.

REVIEWER COMMENTS

Reviewer #1 (Remarks to the Author):

In this study, Murat et al. comprehensively address the occurrence of various mutational processes locally at human origins of replication. Work addressing this type of topic is timely, since the mechanisms generating the mutation rate heterogeneity across the human genome are understudied. A recent publication by the same authors addresses effects of origins on local mutation rates in the germline genome, and here they turn to somatic genomes which enable conveniently studying a wider diversity of mutagenic processes, less confounded by negative selection. They identify that some of the known processes are enriched at origins (UV, AID) and they identify some probably new processes (SBS43 mutations, certain rearrangements – short duplications) which occur apparently exclusively in hotspots. Mechanistic insight is gained via analysis of repair maps, local DNA motifs (AID context, G4) and gene expression analysis for inference of subtypes and cell proliferation. Overall the study has lots of interesting elements, however the various findings reported do need additional support, in our opinion, in form of analyses to rule out various confounders of local mutation rates, replication in additional cancer types, more careful association analyses to gene expression, and considering to include analysis of various other experimental data sets. Please see the remarks below.

Major:

1. A general concern about the study is that there is not clarity to what extent the DNA replication initiation itself, rather than some other process occurring at the same locus, is causal to the changes in mutation rates. There could be better controls for known confounders. It would be important to provide better evidence for this, since causal language is used in the text.

In particular: 1a. regarding the NER deficiency and skin mutations. Can we see evidence that locally lowered NER remains seen at the origins, even when controlling for binding of CTCF and of transcription factors? (in line 150 they actually mention „loading of the pre-replication complex“ as causal, but there was not a test with pre-replication complex binding sites) Does UV mutagenesis at origins remain high after removing the ETS sites?

To address this point, we utilised ChIP-seq datasets generated by the ENCODE consortium, which report binding sites for the CTCF, ETS1, REST, EZH2, and NANOG transcription factors in the GM23338 cell line. We first confirmed the observations of Sabarinathan et al. ¹ that NER activity is reduced at transcription factor binding sites (TFBS) [**Rebuttal Figure 1a**], and that this localised suppression of NER is associated with an increased number of SNVs [**Rebuttal Figure 1b**]. We then examined NER activity at origins that do not specifically overlap with ETS1 binding sites [**Rebuttal Figure 1c**, yellow line], CTCF [**Rebuttal Figure 1c**, green line], or any of the TFBS in the dataset [**Rebuttal Figure 1c**, blue line]. We found that the presence of TFBS within origin domains does not alter the profile of NER activity at origins. We would also like to highlight that transcription factors impose a ~100 nt mutational footprint, which is an order of magnitude smaller than origin footprints we describe. This analysis demonstrates that inhibition of NER at replication origins occurs independently of transcription factor binding.

Rebuttal Figure 1. Contribution of transcription factor binding to NER inhibition at constitutive origins. **a**, NER activity is inhibited at transcription factor binding sites, exemplified specifically by ETS1 and CTCF. All TF = all transcription factor binding sites in the Encode ‘TF clusters’ dataset. **b**, SNVs are enriched at TFBS. **c**, Effect of TF binding on NER activity at constitutive origins.

Unfortunately, no high-resolution datasets of pre-replication complex binding sites are currently available to assess the positioning of pre-replication complexes positioning. Potentially relevant datasets, such as ORC ChIP-seq in human cells², are of insufficient quality and resolution to support any meaningful conclusions (even the ORC1 and ORC2 datasets fail to overlap in this study).

We have included the new data above as **Extended Data Figures 4d, 4e** and **4f**.

1b. Regarding the widespread mutagenesis in Cluster 2 containing various cancer types. G4s are proposed as causal to this mutagenesis. One way to substantiate this is to monitor if mutation rates at origins become unremarkable upon excluding the G4 regions. (see also comment 3b below). Alternatively or in addition, controlling for other possible confounding factors at origins would help.

To address this point, we calculated the mutational burden at origins in cluster 2 tumour samples, with and without the inclusion of G4-forming sequences. We found that excluding G4s reduced the mutational burden by approximately 30% when averaging the signal across all origins and samples (**Rebuttal Figure 2a**). When analysing individual samples, we observed that G4-forming sequences contribute, on average, around 33% and up to 72% to mutagenesis at origins in cluster 2 samples (**Rebuttal Figure 2b**). Thus, G4s significantly contribute to mutagenesis at origins in cluster 2 samples.

Rebuttal Figure 2. Mutational burden at constitutive origins with and without G4s. **a**, Total mutational burden averaged across all cluster 2 samples, with and without the inclusion of G4-forming sequences. **b**, Contribution of G4s to mutational burden on a per-sample basis in cluster 2. G4 contributions were computed as the percentage of mutations occurring within G4 structures.

We have added this data in **Extended Data Figures 5h** and **5i**.

2. The set of origins studied (iniseq2 from a bladder cell line) may or may not be well representative; the different techniques to identify origins have different biases, and as they mention in the discussion there would also be some differences between tissues and between healthy and transformed cells.

The replication origins we have identified with ini-seq represent a relatively small subset of the total potential origins in cells, but they are the most efficient and recurrently utilised sites in the human genome across all cell types examined to date, including both transformed and untransformed cells, as well as the germline^{3,4}. In other words, these origins are not tissue-specific, allowing mutational footprints associated with their presence to be detected above background noise. They also significantly overlap with the core SNS-seq set identified by Akerman et al.^{3,5}. The mutagenic processes we have identified, both in the germline⁴ and in this study, may also occur at other origins, including those that are tissue-specific or used less frequently. However, below a certain threshold of usage frequency, the mutagenic impact of an origin is likely to be obscured by background mutagenic processes.

2a. While this is addressed textually in the discussion, it would be better addressed with actual data. Does using e.g. consensus SNS-seq origins (or another dataset they find appropriate) give similar results?

Ini-seq origins show a highly significant overlap with the SNS core origins identified by Marcel Méchali's group⁵, although the latter dataset is larger³. In both the current manuscript and our previous work on germline mutations⁴, we define SNS-core origins as those identified by Akerman et al., *excluding* those that are also Ini-seq 2 constitutive origins. Therefore, the retrospectively identified most efficient SNS-seq origins will yield the same results as Ini-seq 2, as they occupy the same genomic locations.

We have assessed the mutation rate at SNS-core origins that are not also Ini-seq 2 constitutive origins and found it to be significantly lower (**Rebuttal Figure 3**). (Note: this data is plotted using the same scale as **Fig. 1a** for comparison). This indicates that the mutation rate in cancer genomes is higher at constitutive origins.

Other datasets, such as Ok-seq, bubble-seq, or optical replication mapping, lack the resolution to define specific origins, as they only identify larger initiation zones³, making them unsuitable for determining focal mutagenesis at origins.

Rebuttal Figure 3. Mutagenesis at SNS-core origins that are not also constitutive origins identified by Ini-seq2. The data plotted is for core origins⁵ remaining after those also identified as constitutive³ are removed.

We have added this data to **Extended Data Figure 1g** and discussed this point in the manuscript.

2b. Also, regarding excluding clustered origins; we understand the rationale for doing that but have a slight concern this could introduce a bias, since the clustered origins were said to have distinct biological properties.

This is a valid point. As this reviewer notes, excluding clusters is crucial to accurately assess mutagenesis at origin sites compared to flanking regions. To determine whether clustered origins behave differently from isolated origins, we calculated mutational burden within origin clusters. Specifically, we examined clusters where origins are within 10 kb of each other. We observed an increase in mutation frequency above background levels (**Rebuttal Figure 4a**), supporting the mutagenic properties of both isolated and clustered origins. Next, we calculated the number of SNVs at isolated versus clustered origins and found no significant difference (**Rebuttal Figure 4b**). Therefore, excluding clustered origins from our analysis ensures an accurate comparison of mutation rates to background without introducing bias.

Rebuttal Figure 4. Mutagenesis at clustered vs isolated origins. a, Mutagenesis at origins clustered within 10kb of each other. The median cluster size was 36kb with an average of 7 origins per cluster. The cluster edges are defined by the extreme origins, hence the spikes in mutation at the edges of the ‘meta clusters’ shown in the figure. **b, SNVs count at isolated vs clustered origins.**

We have added this data to **Extended Data Figures 1c** and **1d**.

3. Regarding using pancreatic cancer to study the association between origin-focused mutagenesis, and gene expression programs. A good justification of why only this cancer type is analyzed is lacking. Even though PACA is one of the cancer types with high variability in mutagenesis and a high number of samples, there are other cancer types such as BRCA with similar characteristics and for which it is known that there is a subtype division.

3a. Would it be possible to perform this subtype analysis in one or more other cancer types with enough sample size and report their enrichment and GSEA in supplementary data? In particular Whatever associations they found in PACA e.g with Cyclin D gene expression, or certain pathways, would need to be shown in at least 1 other cancer type, or in a pan-cancer analysis, to be plausible. It seems unlikely that the SBS43-like origin mutagenesis that they identified in Cluster 2 (which contains many tissues) would be confined only to pancreatic cells.

We appreciate this point. We have made significant efforts to extend and corroborate our observations in PACA by performing meaningful analyses in other cancer types or pan-cancer. However, there is very limited matched transcriptomic data for the mutation data in other tumour types within cluster 2, aside from PACA. In fact, the only other cancer type with available transcriptomic data within cluster 2 is ovarian cancer (OV), and only two samples from this matched cohort show a sufficient number of SNVs at origins (> 50) to confidently identify the cluster 2 signature. Any differential gene expression analysis from these samples is thus too noisy to draw reliable conclusions. It is also important to note that BRCA samples with distinct cluster 2 signatures at origins do not have associated transcriptomic data. Nonetheless, we believe the PACA data is valuable, and that removing it entirely would be a loss to the study. We have now tempered the discussion of this data and explained the limitations of this aspect of the study on pages 10 - 12. Further exploration will require matched data of a similar depth to what is available for PACA, and unfortunately, generating such large patient datasets is beyond our capacity.

3b. The more fine-grained analyses of gene expression (meaning: beyond looking at the broad clusters, and focussing on specific pathways or individual genes), like those described on line 335, have potential to be confounded by sample purity, and by aneuploidies. If they want to draw firmer conclusions from this expression correlations (e.g. for glycolysis and pentose phosphate pathways lines 310-326; or downregulation of DNA ss break processing factors lines 335-340), controls for these confounding factors would be warranted. Else these claims would be better marked as speculative, and relegated to supplementary text or discussion or dropped altogether.

The focus of this section of our manuscript is to examine the correlation between the mutagenesis observed at origins in cluster 2 and changes in the expression of pathways known to promote replication stress. The fact that these pathways are dysregulated is sufficient for our analysis and the mechanisms underlying this dysregulation are not relevant to the main focus of the paper. While we acknowledge that aneuploidy may contribute to some of the differences in gene expression we observe, it does not affect our conclusion regarding the link between the upregulation of replication stress-associated pathways and mutagenesis at cluster 2 origins. Additionally, we cannot control for sample purity, which is a general limitation of this type of project.

3c. In addition, it is known that many mutational signatures are tissue-specific to some extent. In the clustering step, would they get the same signatures if clustering the samples according to the 5 clusters but also by cancer type? Finding the same signature (i.e. cluster 2 signature) in several cancer types will show that it is robust.

To address this question, we computed origin mutational signatures in tumours from cluster 2, categorised by cancer type (see below). We found that the characteristic T>G transversions associated with G4 mutagenesis are the predominant mutations across all cancer types.

Rebuttal Figure 5. Origin mutation signatures in individual tumour types contributing to Cluster 2.

We have added this data to the manuscript as **Extended Data Figures 5b**.

By eye it seems that there are subclusters within cluster 5. Have the authors tried to split it?

Indeed, we noticed this as well. While it is possible to deconvolve five subclusters within cluster 5 (**Rebuttal Figure 6a**), the mutational signatures associated with each subcluster are either flat or lack sufficient information (**Rebuttal Figure 6b**). Consequently, we decided not to pursue further examination of these subclusters.

Rebuttal Figure 6. Deconvolution of Cluster 5. a, Cosine similarity heat map separating cluster 5 into four subclusters (clusters 5-1 through 5-4). b, Mutations signatures extracted from each cluster 5 subcluster.

We have added these panels as **Extended Data Figures 3b** and **3c**.

4. To support mechanisms, DNA sequence analysis is used as a proxy for comparing to experimental data, even when the experimental data is available. Two examples of this:

4a. Regarding AID, the tetranucleotide enrichment is fine, but much stronger evidence that oris are AID off-targets would be simply to check overlap with some of the available experimental datasets.

Although there are studies investigating mutations in cancer at preferred AID targeting hotspots^{6,7}, we are not aware of any suitable ChIP-seq (or equivalent) datasets for AID chromatin localisation in human cells, despite a few existing datasets in mice⁸. This may be due to the poor performance of anti-human AID antibodies and the weak nuclear localisation of AID. We are therefore unable to address this point.

4b. Regarding G-quadruplexes: they can indeed be predicted from sequence using a pattern-matching approach, however they were measured experimentally as well (wherein some measurements point to examples that would not be seen with the pattern matching approach).

To gain a deeper understanding of the G-quadruplex motifs that trigger mutagenesis at origins, we analysed the G4 structures identified experimentally using the G4access technique⁹. By examining G4access peaks aggregated from HaCaT, K562, and Raji cells, we characterised experimentally determined G4s within origin domains and found that mutated G4s are predicted to be more stable, according to G4Hunter¹⁰, than all G4s identified at origins [**Rebuttal Figure 7a**]. We then classified G4 subtypes [**Rebuttal Figure 7b**] and discovered that mutated G4s are enriched for short loops and depleted in bulges in tetrads [**Rebuttal Figure 7c**] indicating preferences for more stable G4s. These findings support the notion that only the most stable G4 structures stall polymerase δ and induce mutagenesis at origins.

Rebuttal Figure 7. Features of mutated origin G4s. *a*, Stability of all and only mutated origin G4s predicted by G4Hunter¹⁰. *** $p < 0.001$, Kolmogorov-Smirnov test. *b*, Predicted structural features of all and mutated origin G4s. *c*, Relative motif enrichment in mutated origin G4s compared with all origin G4s.

We have added these panels as **Extended Data Figures 5c, 5d and 5e**.

5. Regarding the methodology, the authors background-adjust the signatures to spot mutational signatures at the origins location specifically. We understand that they need to do this because they extracted the signatures by clustering the profiles directly. However, if a factorization method (such as NMF) is used to extract signatures, the algorithm should directly separate if more than one process is affecting the mutational landscape at origins into different components, possibly reducing the need to background-adjust the profiles according to the flanking regions. Could the authors clarify/explain why they used this clustering approach rather than a more standard approach as it will be applying NMF and state the advantages and caveats in the text.

The cancer datasets analysed in this study are far more complex and heterogeneous than the germline SNP datasets we explored in our previous work ⁴, in which we did identify mutational signatures using non-negative matrix factorisation. We also aimed to highlight mutational patterns present across the genome but particularly enriched at origins. Clustering allowed us to group mutation patterns and their exposures together, rather than analysing them separately, and enabled us to focus on patterns more prevalent at origins without first needing to independently determine the likely number of distinct patterns involved. We have clarified the rationale for our approach in the text on page 4.

Minor:

- Linking the origin-specific signatures activity to external/internal mutagenic exposures may be informative of mechanisms of some of the enigmatic processes occurring at origins. This is related with line 86. „These observations suggest that increased mutation rates at origins are driven by mutational processes of varying degrees of intensity rather than inherent tumour properties“ and line 119 „We then compared these origin-associated signatures with genome-wide signatures derived from samples within the respective clusters (Extended Data Fig. 2b).“ Is there an analysis that compares the activities („exposures“) of known SBS signatures genome-wide with the origin-specific signatures, across tumor samples in each cluster? Ext Fig 2b is related but I think not quite that.

We are not entirely sure what specific experiment the referee is suggesting here. **Extended Data Figure 2a** presents a comparison of the origin signatures from clusters 1 to 5 with the genome-wide COSMIC SBS signatures. When an SBS signature is linked to an underlying mutagenic process, this enables us to infer potential processes that may be active at origins. In other words, the genome-wide SBS signatures do provide insights into processes concentrated at origins, as seen, for example, in clusters 1 and 4.

- Lines 75-76 „we corrected the apparent mutation rates of each of the six pyrimidines mutation by their occurrences and found increased mutation rates associated to each type (Fig. 1a).“ I presume this is control by trinucleotides? If not, it should be performed; if it is, then make it clear in the text.

Our originally presented data were corrected for local variations in base composition but did not account for trinucleotide context. We have now made this adjustment and observed an increased prevalence of mutations for all pyrimidine mutations except C>T [**Rebuttal Figure 8a**]. Additionally, we separated C>T mutations into those occurring in CpG and non-CpG contexts, finding that C>T mutations in a CpG context are the only ones not enriched at origins [**Rebuttal Figure 8b**]. This mirrors our observations in germline mutagenesis, where mutational processes at origins overshadow the background clock-like SBS1 signature – SBS C in our germline origin mutation analysis ⁴ – and/or factors at origins protect against spontaneous and/or enzymatic cytosine deamination.

Rebuttal Figure 8. Mutation enrichment at constitutive origins corrected by trinucleotide context. **a**, Mutation frequency by each substitution in the pyrimidine context corrected by trinucleotide frequency. **b**, C>T mutations at origins segmented by whether or not they are in a CpG context.

We have added these panels as **Extended Data Figures 1e** and **1f**.

- Lines 112-116 „Cluster 2 signature exhibits high similarity to SBS43 (cosine similarity = 0.95), a signature of unknown aetiology,“ COSMIC states SBS43 may be an artefact – any comments on this?

We note that SBS43 is listed only as a ‘possible sequencing artefact’ in the COSMIC database. However, it has been identified by multiple sequencing centres using different variant callers and replicated in studies beyond its original discovery (Ref: COSMIC website - <https://cancer.sanger.ac.uk/signatures/sbs/sbs43/>). The COSMIC website also reports that SBS43 is enriched in early replication timing zones but does not show transcriptional or replication strand asymmetry. Crucially, in our analysis, SBS43 is not observed at potential G4s across the genome but is concentrated at origin-associated G4s, and it is not found in all cancer types. These findings are inconsistent with SBS43 being an artefact. We have presented a biologically plausible explanation for this signature, which is further supported by our previous *in vitro* data ¹¹. We have discussed our findings with Ludmil Alexandrov, one of the originators and curators of the COSMIC signature database. He agrees that our analysis offers a potential explanation for SBS43, and it will be included in a future update of the database.

„Cluster 3 signature shares moderate similarities with the known SBS10b signature (cosine similarity = 0.64, associated with polymerase epsilon exonuclease domain mutations).“ 0.64 is very low similarity and so it is not likely to link the mechanism to SBS10, so writing this could be misleading.

We agree that this aside is not sufficiently substantiated and have removed it.

- Line 357 „Taken together, these findings indicate that breaks arising from replication initiation are transformed into a diverse array of short tandem duplications,“ did they consider that these duplications might result from re-replication (where a break, in one interpretation, would not be considered an initiating event). Could merit discussing.

We have expanded the discussion section of the manuscript to detail our rationale.

- Line 418 in Discussion. Could they clarify how does this hypothetical mechanism of switching to another region w/microhomology cause duplications but not deletions?

We take these two interesting points together. The reviewer is correct in noting that we cannot definitively prove a causal link between breaks at origins and the observed local tandem duplications, although it is clear that something occurring at these origin sites results in their enrichment at CNV boundaries.

We have previously demonstrated experimentally that breaks are enriched at origins ⁴, and several mechanisms could contribute to this. One likely mechanism involves topoisomerase activity, with type 2 topoisomerases also being enriched at efficient origin sites ⁴. Paradoxically, the most efficient origins are also often situated within complex, secondary structure-rich sequence contexts ³, where these structures may present immediate obstacles to the nascent polymerase δ -driven replication fork.

Secondary structures, particularly G4s, have been associated with CNV boundaries in both cancer¹² and the germline¹³. Thus, CNVs at origins may also be linked to the enrichment of sequences with structure-forming potential near origins³.

In addition to causing breaks, secondary structures may induce fork stalling and template switching, which can lead to non-allelic homologous recombination between repetitive sequences through a process known as Fork Stalling and Template Switching (FoSTeS)¹⁴, potentially resulting in CNVs. Fork stall/template switch models, as opposed to those driven by frank breaks, tend to favour duplications because the disengaged nascent strand reanneals to an incorrect template via microhomology, generally adding rather than deleting sequence. On the other hand, frank breaks may be processed through classical non-homologous end-joining mechanisms, leading to deletions.

Although the data we present do not allow us to determine the exact mechanism underlying the association between efficient origin sites and CNVs, we have reworded and expanded the discussion of the relationship between different sources of origin-associated break and genetic outcome, as suggested.

- Line 435 in Discussion „intrinsic biology of cancer subtypes significantly influences the intensity of mutagenesis observed at these sites.“ is overstating the impact of the data, if this remains shown only for one cancer type as currently (pancreatic).

See our discussion of point 3a above. We have tempered the wording to acknowledge the limitation of the deeper analysis in only one cancer type.

- How the “Minor allele calculations” are performed seems not explained in the methods.

Details about minor allele frequency calculation have been added to the **Methods** section.

- Line 81 to 84 when they compare the mutation rates at origins they say that cancer types such as gastric and gallbladder have high mut rates based on fig 1b, but when they normalized the mutation rates by the flankings (results in fig 1c) we can see that the gastric cancer GACA (stomach) is below 1 meaning its relative mutation rates its not high. It is confusing that it is used as an example of high mutation rates in the text.

Thanks for pointing this out. In retrospect, GACA was not a good example to choose. While GACA exhibits an increased mutation prevalence and origins compared to flanks when considering overall mutation load across all the samples in this cancer type (**Figure 1b**), when examining each sample in turn, the average ratio is < 1 (Figure 1c). We have reworded this sentence.

Reviewer #2 (Remarks to the Author):

In this work, Murat et al. go forward from their 2022 work focusing upon constitutive origins (COs) in humans to try and find mutational processes at replication origins. They use single nucleotide variants from across the ICGC pancancer cohort, deriving ratios of mutations between origins and their flanking regions binning them into characteristic clusters, before attempting to resolve the mechanisms underlying profiles in some clusters. While there's interesting work here, their analyses are flawed, relying upon poor statistical assertions to knit together too many wide-ranging analyses. Fundamentally, their work is undermined by well-understood alternative explanations that are ignored but lie at the bases of their analyses. It is clearly a work of great effort, but I think a significant change in narrative angle and better delivery, especially in use of statistics guiding the narrative, is needed.

We were disappointed to read the reviewer's comment that **[redacted for confidentiality]** did not find it 'worth going further' which suggests they may not have thoroughly examined the entire manuscript. We respectfully disagree with the statement that 'well-understood explanations' were overlooked or inadequately addressed in our original submission, and we maintain confidence in our narrative and the presentation of our results. We have addressed all of the points raised by the reviewer below. However, we feel compelled to highlight the pejorative tone of many of the comments, which we find inappropriate in the context of a scientific review.

Deriving COs was initially to acquire a set of dependable origins with which to generalise with across tissue types but then to break them down in this way doesn't make sense- yes, the CO come from those in the bladder cancer cell line and the cell lines used by Akerman, but it's highly probable that differences in expression and a permissive chromatin environment impact DNA damage and repair associated with various mutational processes that are highly likely to coincide positionally with stable origins. For example, Conrad Nieduszynski just just this as part of a recent preprint where they propose that transcription tightly confines replication at active genes <https://www.biorxiv.org/content/10.1101/2024.04.28.591325v1.full.pdf>. [\[biorxiv.org\]](https://www.biorxiv.org)

We have previously shown that constitutive origins are closely linked to transcriptional units, a finding that aligns with Marcel Méchali's study on the most frequently used origins identified by SNS-seq⁵. However, constitutive origins are distinct from transcription start sites⁴. The recent study by Conrad Nieduszynski employs an elegant technique; however, it currently lacks the power to draw definitive conclusions, as it is based on a dataset of only approximately 2000 events attributed to origin firing.

We acknowledge that both transcriptional and replication programmes undergo significant changes in many cancers, which may account for the lack of detectable changes in mutational patterns at our constitutive origins in some cancer types—a point we address in the manuscript. This topic merits further exploration in future studies. Nevertheless, constitutive origins have a considerable impact across a wide range of cancer types, and we are confident that our conclusion—that sites of replication initiation can influence mutagenesis in cancers—is robust.

In the 2022 paper, the functionality of included COs is well examined, but is missing here, and while I would expect to be told to refer to that paper, it looks like the set of COs is derived in the same way but is less than half the size.

We are unclear what the reviewer means by 'functionality.' We have previously identified a small subset of potential replication origins that are conserved across multiple cell types, including the germline, and are used frequently enough to influence local mutation rates. Our data demonstrates for the first time that these same origin sites can also affect mutational processes specific to certain cancer

subtypes. Regarding the number of constitutive origins examined in this study, our set remains consistent with those we described previously, consisting of ‘isolated’ origins, as we have excluded clustered origins (see discussion of point 2b from reviewer #1).

A big red flag associated with this is that the cancer sampling clusters in Fig 2a are only apparent in very particular cancer types. Cluster 1 is driven by UV damage, obviously, and cluster 4 is something particular to lymphoma (MALY).

We do not view the observation that clustering has identified specific genome-wide mutational processes modulated by origins as problematic. This finding is anticipated, as we designed the clustering to detect mutation patterns at origins, and it successfully identifies examples that also appear genome-wide. In a subsequent layer of analysis, we examine the extent to which these mutational patterns are concentrated at constitutive origins. It is not surprising that AID-associated mutagenesis is observed in germinal centre B cell lymphomas, as this is the cell type in which AID is expressed. However, it is noteworthy that this mutational pattern has not been previously demonstrated to be focused on sites of replication initiation.

Though cluster 2 is composed of multiple cancer types, it could be driven by one or two cancer types.

See above for our response to point 3c of Reviewer 1. The signature identified in cluster 2 is seen in multiple cancer types within that cluster.

Debatably, all of these clusters are driven by a small number of origins. You could do a much better job unpicking this, for example, plotting 1b for samples at specific COs that contribute to specific clusters against their other COs and at COs in other cancers. This would be instead of the way you use mutation spectra, which has created a biased analysis. Presenting the data from 1a, such as providing the CO counts (or even as bed files) and sample counts in a table would be useful to assess this. Importantly, regression would be an easy and powerful way of asserting that the enrichment of mutations at origins is driven by particular cancer types or their properties, such as the data in Extended Data Figure 1d, where total genomic burden is plotted against the ratio of mutations at origins against their flanks and gives a really strange distribution.

We interpret the reviewer’s comment as suggesting that a bias in our analysis may arise from a small and recurrent number of origins that are overrepresented within the set of origins exhibiting mutations. To address this concern, we specifically evaluated the number of origins contributing to mutagenesis in each cluster. First, we assessed the number of origins with at least one SNV per sample and found a broad frequency distribution [**Rebuttal Figure 9a**]. Next, we examined the number of contributing origins per sample and per cluster [**Rebuttal Figure 9b**]. Notably, cluster 1 (MALY) is composed of a larger number of contributing origins compared to the other clusters. However, UMAP projections of cancer samples based on the contributing origins to mutagenesis—coloured by either cancer type [**Rebuttal Figure 9c**] or origin mutation type cluster [**Rebuttal Figure 9d**]—do not reveal any clustering. Therefore, we conclude that no specific subset of origins contributes disproportionately to our mutational signature analysis, and there is no bias related to origin usage.

Rebuttal Figure 9. Number of origins contributing to mutagenesis in each cluster. **a**, Distribution of the number of origins in each sample that have at least one SNV. **b**, Number of origins contributing to each mutation type cluster. UMAP projections of cancer samples based on origins contributing to mutagenesis, coloured by **(c)** cancer type and **(d)** mutation type cluster.

This could also improve how you pull apart the data to make assertions, which you occasionally make with quite dubious correlation coefficients. For example, at L90 you demonstrate a strong correlation between the total number of mutations and those at origin, where $R=0.954$, but then go on to say that the origin/flank ratio is the opposite of this, when $R=-0.246$, is not a fair comparison. Your reported correlation coefficients from L152 are not justifiably extreme enough to make the assertions about NER that you make: 0.4, 0.17, -0.44 and -0.36. Relating to this, on L85 and L128 you say differences are significant, but there are no associated statistics. In association with this, I don't like that the KS test is used when a Mann-Whitney is more appropriate. The KS is also less conservative and more likely to give a significant P value.

The correlations referred to by this reviewer are reported to indicate the direction of the correlation, which in this context has some important biological consequences. It is unclear what would constitute a 'fair comparison' or 'extreme enough' to support our assertions. The word 'significant' has been removed when describing qualitative observations.

Regarding the choice of KS over Mann-Whitney tests: Our manuscript focuses on comparing the distributions of values (e.g., SNV and CNV counts, G4 stability scores) and evaluates the null hypothesis that the two observations come from the same distribution. The KS test, which calculates the maximum distance between the empirical cumulative distribution functions, is suitable for this

purpose. While we acknowledge that KS tests may be less conservative, their sensitivity to differences in both location and shape between distributions is important in this context.

Another line of evidence propping up the analysis in the early stages is use of mutational spectra, though this is deployed clumsily. For example, in L122, you say that the lack of resemblance to extant signatures validates your approach to looking for mutagenic processes at origins, but the set up is biased in a way that will fail to resemble known signatures. This is demonstrated in EDF 2b, which I actively dislike as, of course multiple signatures packed on top of each other, consisting of multiple mutational processes, will generate weird signatures, even if they're across the same sorts of genomic feature. To do this properly in the future, you should consider de novo signature deconvolution and fitting.

In our previous analysis of germline mutagenesis⁴, we could reasonably assume a limited number of individual mutational processes. Consequently, we employed a signature deconvolution approach using non-negative matrix factorisation to analyse mutations at and around origins. However, given the extreme complexity of mutagenesis in cancer genomes, we adopted a different strategy. We used clustering to group cancers with similar mutational processes at origins and then extracted signatures from these clusters. We assessed how each cluster signature correlates with known COSMIC SBS and its focus on origins. While Clusters 3 and 5 consist of multiple processes, Clusters 1, 2, and 4 exhibit strong cosine similarity to existing COSMIC signatures, as discussed. We are confident that our approach is valid and has produced biologically significant findings.

It's worth noting that, similar to my previous statements about correlation, on L115 you cite a cosine similarity of 0.64, but this is too low to be considered similar to the "matching" COSMIC signature.

As discussed above in response to minor point 4 of Reviewer 1, we agree that this speculation is on insufficiently firm ground and have removed it in the revised version.

Another concern is that, on L112 you consider SBS43 to be relevant, though this is a sequencing artefact that isn't included in COSMIC or the PCAWG papers.

We do indeed consider SBS43 to be relevant and have discussed it with one of the curators of the COSMIC database. We elaborate on this point above in response to minor point 3 of Reviewer 1.

To go back to other mechanisms that potentially explain the mutational patterns these authors are seeing, I'll focus on cluster 1. It's well known that DNA damage from UV is occluded by bound transcription factors and therefore has elevated mutation rates. While you acknowledge this in L149, you don't do anything to demonstrate that the COs responsible aren't at transcription start sites of genes expressed in the relevant cell of origin.

We have shown previously that while constitutive origins significantly overlap with transcriptional units and can be found in the vicinity of transcription start sites across all tissue types, they are distinct objects⁴.

In addition, there isn't statistical analysis to assert that NER activity is different between the leading and lagging strands. This finding is odd, given that it was generally accepted that there were lower mutation rates on the leading strand, and more recent evidence has demonstrated that this effect is the result of biases in expression. Nuria Lopez-Bigas' group has performed relevant analyses and has nicely demonstrated how to effectively compare against expected values that would strengthen the

analyses here: <https://www.sciencedirect.com/science/article/pii/S009286741930234X>[[sciencedirect.com](https://www.sciencedirect.com)]

The difference between leading and lagging strand mutation rates remains a matter for debate.

[redacted for confidentiality] Most determinations of mutational asymmetry in vertebrate cells do not consider the immediate vicinity of origins, in part because sites of lack of precision with which efficient initiation sites could be identified. As we have discussed previously ⁴, and in the present manuscript, mutagenesis in the immediate flanks of origins may differ from other regions of the genome not least because of differences in the polymerase composition of the replisome. Quantification of the differences in NER activity on the lagging and leading strands is provided in **Extended Data Fig. 4h**.

I don't deny that the other analyses have merit, but given the poor bases from which these are formed, it's not worth going further. I think it would be better to frame this paper as the distinction between aspects of certain COs and proving that elevated mutation rates are not just incidental to other damage and repair processes. It would also be useful to try and expand your analyses into a few cell lines with consistent origins in early replicating regions. For example Petryk et al (<https://www.nature.com/articles/s41586-020-1943-3> [[nature.com](https://www.nature.com)]) report ok-seq from two reps of B cells and there's plenty of WGS from lymphomas available, all highly related to your MALY story.

We find it difficult to fully grasp the reviewer's comments and suggestions here. As noted above, we are happy with the way in which we frame our results. It is unclear how we could apply our work to "a few cell lines" using OK-seq data, which, as discussed above, is designed to focus on initiation and termination zones rather than mapping replication origins. Ok-seq simply does not have the resolution for the analyses we present. The connection between the cited manuscript (Alexandrov et al. (2020)) and this point is not apparent, but we presume the link was meant to be to Petryk et al. Nat Comms 2016.

I truly hope to see a much improved version of this work in the future.

[redacted for confidentiality]

I've included other notes that I made going through the manuscript, which I hope are useful:
The number of origins under consideration isn't always clear and might be insightful. A proper definition of constitutive origins is needed in the main body of the text.
L68 It would be excellent to include an approximate description of the sites you're looking at initially-how big, how many and how conserved, though this latter point could be included in the introduction.

We provided a detailed explanation of the analysed origins on pages 15 – 16 of the original submission (as part of the online methods). This is called out at the start of the results.

L75 It's standard for the field, but unclear to the uninitiated that these mutation classes include their complements, e.g. T>A also represents A>T.

We believe this is obvious but have clarified the point.

L150 This is also explained by TF binding of adducts, occluding repair.

We are not aware of any reports demonstrating that transcription factors (TFs) can directly bind to UV adducts. We believe the reviewer is referring to the concept that UV lesions may influence the specificity of TF binding to their consensus sequences and/or create new binding sites by inducing local change in DNA helix structure (refer to Mielko et al., PNAS 2023 as example ¹⁶). However, it would be challenging to assess this without knowing which specific transcription factors and binding sites are affected. Importantly, our analysis shows that accounting for known TF binding sites does not alter our conclusion regarding the reduction of NER at origin domains (see Reviewer #1, point 1).

L214 stem, not stems

Corrected

L256-258 reference needed

The relevant references are present, just earlier in the sentence.

L265 Why these tumours? What links them? Pulling apart effects by cancer type will not only get towards a mechanism.

As discussed above, further detailed dissection of cluster 2 in the manner we have done for the PACA dataset is not possible due to lack of extensive matched mutation and transcriptomic data for other cluster 2 cancer types.

PACA story - 8oxoG would likely be enriched at G4 sites. Can you discount that there is an enrichment at COs compared to other parts of the genome? Is G4 formation enriched at COs compared to other origins and the rest of the genome?

The point about oxidation of G at G4s is not one that we consider to be of central relevance to the current study and we have not investigated it. We have explored the features of G4 at origins that are associated with mutagenesis, as discussed above and this data is incorporated in the new manuscript in **Extended Data Fig. 5c, 5d and 5e**.

Figures general statements: I would smooth your line plots to make it easier to see the profile of effects, e.g. 2d and EDF 2C. You've got error bars on lots of plots, but I don't know what these mean.

We believe smoothing is not appropriate in this case and generally avoid it to prevent misrepresentation of the true signal. Displaying the noise in the data highlights which associations are genuinely robust and which are not.

Error bars are defined in all cases. In cases where multiple instances of the same type of graph appear in a Figure (e.g. box plots) this is done at the end of the relevant legend.

Fig 1 legend Add fold change (FC)

Done

Stats for 2e and f would be nice, to see if they're significantly enriched.

These analyses are not intended to compare clusters directly, but rather to display the distributions of SNV counts and mutational burden at origins within each cluster. Enrichment is evaluated in separate experiments.

Fig4b and c, in other plots the points are overlaid on top of box plots, the number of origins in each category would at least be useful

Figures 4b and 4c compare the predicted stability of G4 structures with and without mutations, without taking specific origin categories into account.

Fig4e How do you get negative contribution from SBS10c?

There are no negative values in this plot. The scale for POLE (blue) is shown on the left Y-axis, while the scale for POLD1 (red) is on the right Y-axis. This is now clarified in the legend.

EDF1c and d What are the error bars?

This information has now been added.

EDF1d 1c goes up to 5, is the data in EDF1d not the same?

The data shown in **Fig. 1c** and **Extended Data Fig. 1d** are the same. However, Fig. 1c displays individual values, while Extended Data Fig. 1d presents the mean and standard error of the mean.

EDF2a these are in a not very helpful order

We have reordered the signatures.

Fig5d What lies in the extremes of the distribution? A table of them all would be interesting.

Pathways at the extremes of the distribution are challenging to interpret in the context of our work, such as depleted pathways (e.g., KEGG_PROTEASOME, KEGG_AMINOACYL_TRNA_BIOSYNTHESIS, KEGG_PARKINSON'S_DISEASE) and enriched pathways (e.g., KEGG_NEUROACTIVE_LIGAND_RECEPTOR_INTERACTION, KEGG_VASCULAR_SMOOTH_MUSCLE_CONTRACTION, KEGG_MELANOGENESIS). The full details can be accessed through the published code.

Data availability: the link for the code associated with this work doesn't work. I would expect a bed of the processed COs to be available too.

The code is now available, and we have provided a BED file listing the origins included in this study.

Occasionally, clusters are grouped for convenience, rather than the distinction appreciated. For example, Clusters 1, 3 and 5 are plotted together, assumingly to emphasise the peak at the origin in cluster 1, which would be diminished when plotted with clusters 2 and 4.

We have clearly stated that the enrichment of mutations at origins in cluster 1 above the flanks is minimal, as emphasised in **Fig. 2f**. However, this enrichment does exist, and due to the extensive genome-wide exposure of this signature, even this modest increase at origins enables the detection of an enhancement of this signature.

Defining PACA 2 and 3 as distinct and then ignoring that in the analyses looks bad.

We do not ignore this point. We make the distinction quite clear in **Fig. 6** and **Extended Data Fig. 5** and **6**.

Reviewer #2 (Remarks on code availability):

Corrected with apologies.

Reviewer #3 (Remarks to the Author):

The authors explored the role of DNA replication in driving mutagenesis and genomic rearrangements across various human cancers. The study comprehensively analyzed mutation signatures at the consecutive replication origins. They analyzed mutation calls from the International Cancer Genome Consortium aggregated from 86 cancer projects. They found that sites of replication initiation act as hotspots for mutagenesis in human cancers. Subsequently, the authors focused on tumors with over 50 mutations at origins (227 tumors across 17 cancer types) and characterized the mutation patterns around the origins. These origins are associated with increased mutation rates compared to its 2.5kb proximal flanking regions and genome-wide. They identified five mutation patterns enriched at the replication origins, of which clusters 1 and 4 are specific to melanoma and malignant lymphoma, respectively. They found that the aggregate origin signatures derived for clusters 3 and 5 show no discernible contribution to origin mutagenesis, leaving mutation cluster 2 (enriched in pancreatic cancer, esophagus adenocarcinoma, and breast cancer) as a newly identified mutation signature associated with replication initiation.

The authors then conducted a series of analyses linking mutation signature clusters to mutation processes. They found that mutation cluster 1 resembles COSMIC SBS7b, a process linked to UV light and nucleotide excision repair. The author proposed that this resemblance is due to impaired NER at the origin. Next, the authors analyzed the XR-seq generated using human fibroblasts NER accessibility at the replication origins. They found that mutation density at origins correlates negatively with NER activity, suggesting that replication origins inhibit NER (although it remains unclear how TT dimers were transformed into C mutations). The authors then investigated the process for mutation cluster 4, finding this cluster significantly similar to SBS84, which was proposed to depend on AID. They concluded that the replication origins are off-targets of AID. The authors also investigated mutation cluster 2, finding that T to G mutations at the origin occur within G-rich sequences, particularly embedded within G tracks.

This paper demonstrates that each cancer type has distinct mutation signatures around the replication origins. With cluster 2 as a newly identified mutation signature at the origins, enriched in three major cancer types, this manuscript offers new insights into cancer genome evolution.

The reviewer noted one major shortcoming in the current manuscript. Given that the authors analyzed mutations around the constitutive origins, the findings that each tumor type displays different mutation signatures are very interesting. The authors have implied that these are tissue/cell-type specific lesions. Considering that new technologies have significantly improved sequencing accuracy (e.g., Nanoseq), the authors could address the mutation signatures at or around replication origins in the precancerous stage or normal tissues to clarify whether these mutation clusters are cancer-specific. These additional analyses and clarifications would significantly enhance the manuscript's contributions to understanding the complex dynamics of DNA replication, repair mechanisms, and mutagenesis in human cancers.

We thank this reviewer for this suggestion, as demonstrating the lack of mutagenesis at origins in normal tissues supports our observations being specific to cancer biology. We examined datasets from non-proliferating tissues¹⁷ and pre-cancerous somatic tissues¹⁸. Our analysis found no evidence of SNV accumulation (**Rebuttal Figure 10a**) or an increase in specific substitutions (**Rebuttal Figure 10b**) at or around constitutive origins in the Nanoseq datasets from Abascal et al.¹⁷. This supports the notion that the mutations we report are linked to replication. To investigate the impact of origins on mutations in pre-cancerous somatic tissues, we analysed mutations from 29 cell types reported by Moore et al.¹⁸. While we did observe an increase in overall SNV burden at constitutive origins (**Rebuttal Figure 10c**), this can be attributed to local variations in base composition, as there is no evidence of increased mutation rates when corrected for base composition (**Rebuttal Figure 10d and 10e**). This supports the hypothesis that the mutations observed in the cancer datasets arise from cancer-specific

mechanisms and/or unconstrained proliferation. We have added these findings to the manuscript as a new **Extended Data Fig. 2**.

Rebuttal Figure 10. Origin mutagenesis in non-proliferating and pre-cancerous somatic tissues.

a, Total SNVs in the vicinity of constitutive origins as determined by Nanoseq in non-proliferating tissues (Abascal et al., 2021). **b**, Specific SNV substitution patterns surrounding constitutive origins identified by Nanoseq in non-proliferating tissues (Abascal et al., 2021). **c**, SNV burden around constitutive origins in non-cancerous somatic tissues (Moore et al., 2021), not adjusted for local base composition. **d** and **e**, SNV base substitution rates around constitutive origins, corrected for base composition.

In addition, the author also found that replication origins are hotspots for genome rearrangement. This notion has been mentioned previously by Barlow et al. (<https://doi.org/10.1016/j.cell.2013.01.006> [doi.org]) and Macheret et al. (<https://doi.org/10.1038/nature25507> [doi.org]), where early replicating replication initiation sites are hotspots for DNA breaks and genome rearrangement. The author should further specify how their contribution differs from previous publications.

Both studies suggest that sites of replication initiation are also sites of damage and potential genome rearrangement. In the case of Barlow et al., these are locations of physiologically initiated replication in B cells, while Macheret et al. investigate ectopic sites of replication initiation induced by oncogene overexpression. However, the resolution with which sites of replication initiation are inferred—through RPA accumulation following HU treatment in Barlow et al. and EdU-seq in Macheret et al.—is significantly lower than that of our ini-seq2 constitutive origin dataset. Neither study specifically examines the patterns of SNV mutagenesis around sites of replication initiation, which is the primary focus of our work, and indeed neither technique has the resolution to facilitate such an analysis. Nevertheless, we acknowledge that both papers are important and relevant in the context of our findings that origins are hotspots for genome rearrangements, and we have now cited them.

Other comments:

1. The authors have indicated that nucleotide excision repair (NER) activity is generally suppressed at the origin of replication. It would be beneficial if the authors could speculate on potential reasons for this suppression.

We have expanded our discussion on this point to explicitly speculate that the suppression of NER at origins is due to the binding of the pre-replication complex (pre-RC) and is mechanistically similar to the suppression of NER at transcription factor binding sites ¹. As mentioned earlier in our response to point 1a from Referee 1, it is currently not feasible to directly test this hypothesis due to the absence of high-resolution maps of ORC binding on chromatin.

Additionally, considering other mechanisms that could contribute to the C to T mutation beyond NER would enrich the discussion.

Please refer to the response to Referee 1 above. As the referee will recognise, C to T mutations can arise from various mechanisms targeting both C and G. A significant genome-wide source is the spontaneous deamination of methylated cytosine, which is believed to contribute to COSMIC signature 1, a widespread background 'clock-like' mutational signature (Alexandrov et al., 2015). Our further analysis of C to T mutations in their trinucleotide context reveals that specifically C to T mutations within a CpG context are reduced at origins. In our previous analysis of germline mutations at origins (Murat et al., 2022), we showed that this signature (mapped to SBSA in our analysis) accounted for most of the mutagenesis identified around origins. Interestingly, this signature is suppressed at the origin sites themselves, despite the increased GC content (Guilbaud et al., 2022). We speculated that this background signature is 'overwritten' by more dominant mutational processes at highly active origins. Alternatively, it may indicate a protective effect against spontaneous deamination or other DNA-damaging processes due to pre-RC binding. We have revised and expanded our discussion of this point on page 3 of the revised manuscript.

The manuscript also mentions activity at the lagging strands near the origins. Clarification on whether this is a property specific to lagging strands or unique to replication initiation zones would be valuable.

This point is difficult to assess. As shown in **Rebuttal Figure 11**, the observed strand NER asymmetry at origins decreases at approximately 5 kb from the origin midpoints. However, it is unclear whether this suggests that strand asymmetry is limited to the origins or if it is due to the loss of signal phasing at these sites. To answer this question more definitively, it would be necessary to incorporate data from replication timing and/or OK-seq phased at origins, along with XR-seq datasets. Unfortunately, these datasets are currently unavailable, so we cannot draw a conclusion on this point.

Rebuttal Figure 10. NER-corrected activity strand bias at origins. This plot is analogous to Extended Fig. 4h but features extended scales.

2. Regarding the AID-dependent mutation signature identified in cluster four, it is noted that AID is predominantly expressed in B cells. This raises questions about the presence and implications of such mutations in both B and T cell lymphomas.

It is important to highlight that the MALY cancer set is exclusively derived from B cells (**Rebuttal Table 1**), specifically from germinal centres. This set includes the exact type of lymphoma likely to express AID. The metadata for the project from ICGC is provided below. We have clarified that these are B cell malignant lymphomas in the text.

Code	MALY-DE
Name	Malignant Lymphoma - DE
Primary Site	Blood
Tumour Type	Malignant Lymphoma
Tumour Subtype	Germinal center B-cell derived lymphomas
Countries	• Germany
Number of donors in PCAWG	• 101
Number of donors with molecular data in DCC	252
Total number of donors	274

Rebuttal Table 1. Metadata for the MALY project on ICGC.

Given that approximately 50 off-target sites for AID have been identified in mouse B cells (<https://doi.org/10.1016/j.cell.2011.07.049> [doi.org]), which are hotspots for translocations, a comparison of AID's contribution at the origins to other AID off-target sites in B cell lymphoma, genome-wide, would be insightful. Furthermore, the base excision repair pathway removes deaminated cytidine at AID targeting sites, leading to DNA double-strand breaks and potential translocations. It would be valuable if the authors investigate whether more translocations are observed around the origins compared to the rest AID off-target sites in the genomes.

This is an interesting idea; however, it is not currently feasible to investigate it. As the referee notes, the datasets in Chiarle et al.¹⁹ are derived from mice, and we are not aware of any equivalent dataset in human cells. Additionally, we do not yet have ini-seq2 data available for mouse cells.

3. The authors conclude that increased mutation rates at origins may result from replicative stress induced by excessive cell proliferation based on GSEA analysis findings related to the MAPK and JAK-STAT pathways. However, these pathways are not exclusively involved in regulating cell growth, DNA replication speed, or the number of additional origins fired in cells. To strengthen the assertion that replication stress or cell proliferation causes the mutation cluster type at the origin, the authors are encouraged to explore other datasets that directly test mutation density under experimentally controlled conditions altering cell growth/proliferation. Examples of such conditions in normal cells include sonic hedgehog activation in granule neuron progenitors (<https://doi.org/10.1038/s43018-020-0094-7> [doi.org]), and erythroid progenitors (<https://doi.org/10.1126/sciadv.1700298> [doi.org]).

This is a conceptually appealing idea, and both would be interesting systems in which to test it. However, neither of these papers includes any sequencing data reporting induced SNVs, making the proposed analysis unfeasible

4. Regarding methodology and details in analytic approaches, the authors should, perhaps in supplementary information, provide detailed versions of R and packages. In line 581, please describe “a standardised protocol” to clarify the rationale and specifications for readers who have not done this type of analysis before.

A comprehensive set of R notebooks is available on GitHub, offering annotated code that explains how each figure was created. We apologise for the broken link at the time of submission.

5. Across the manuscript, the “online methods” were mentioned several times while the reviewers did not have access to this method section. Please make sure this information is available.

The ‘online methods’ were present in the submitted manuscript on pages 15 – 20.

6. At “Data availability,” the link to <https://github.com/Sale-lab/OriCan> [github.com] was not found.

Apologies that the link did not work. We had inadvertently not set the public access tab correctly. This is now corrected.

7. Figure 1, legend: please describe what “FC” stands for.

We have spelled out ‘fold change’ in the Y-axis label of Figure 1b.

8. Figure 2e, legend: please explain the Y axis “log10 scale”. Is the scale in log10, or does it mean something else?

Yes, this means the axis is plotted on a log10 scale.

9. Figure 3, legend: please spell out the full name for “CPF” and “PP” in the legend.

We believe the referee means ‘CPD’ and ‘PP’. These are acronyms for cyclobutane pyrimidine dimers and 6–4 pyrimidine–pyrimidone photoproducts respectively, the two major DNA lesions induced by UV light. This is now clarified.

Reviewer #3 (Remarks on code availability):

The link to the code (<https://github.com/Sale-lab/OriCan> [github.com]) cannot be found.

Apologies, as above. This is now fixed.

Reviewer #4 (Remarks to the Author):

References

1. Sabarinathan, R., Mularoni, L., Deu-Pons, J., Gonzalez-Perez, A. & López-Bigas, N. Nucleotide excision repair is impaired by binding of transcription factors to DNA. *Nature* **532**, 264-267 (2016).
 2. Tian, M. et al. Integrative analysis of DNA replication origins and ORC-/MCM-binding sites in human cells reveals a lack of overlap. *Elife* **12**, RP89548 (2024).
 3. Guilbaud, G. et al. Determination of human DNA replication origin position and efficiency reveals principles of initiation zone organisation. *Nucleic Acids Res* **50**, 7436-7450 (2022).
 4. Murat, P. et al. DNA replication initiation shapes the mutational landscape and expression of the human genome. *Sci Adv* **8**, eadd3686 (2022).
 5. Akerman, I. et al. A predictable conserved DNA base composition signature defines human core DNA replication origins. *Nat Commun* **11**, 4826 (2020).
 6. Benitez-Cantos, M. S., Cano, C., Cuadros, M. & Medina, P. P. Activation-induced cytidine deaminase causes recurrent splicing mutations in diffuse large B-cell lymphoma. *Mol Cancer* **23**, 42 (2024).
 7. Hernández-Verdin, I. et al. Pan-cancer landscape of AID-related mutations, composite mutations, and their potential role in the ICI response. *NPJ Precis Oncol* **6**, 89 (2022).
 8. Álvarez-Prado, Á. F. et al. A broad atlas of somatic hypermutation allows prediction of activation-induced deaminase targets. *J Exp Med* **215**, 761-771 (2018).
 9. Esnault, C. et al. G4access identifies G-quadruplexes and their associations with open chromatin and imprinting control regions. *Nat Genet* (2023).
 10. Bedrat, A., Lacroix, L. & Mergny, J. L. Re-evaluation of G-quadruplex propensity with G4Hunter. *Nucleic Acids Res* **44**, 1746-1759 (2016).
 11. Murat, P., Guilbaud, G. & Sale, J. E. DNA polymerase stalling at structured DNA constrains the expansion of short tandem repeats. *Genome Biol* **21**, 209 (2020).
 12. De, S. & Michor, F. DNA secondary structures and epigenetic determinants of cancer genome evolution. *Nat Struct Mol Biol* **18**, 950-955 (2011).
 13. Bose, P., Hermetz, K. E., Conneely, K. N. & Rudd, M. K. Tandem repeats and G-rich sequences are enriched at human CNV breakpoints. *PLoS One* **9**, e101607 (2014).
 14. Lee, J. A., Carvalho, C. M. & Lupski, J. R. A DNA replication mechanism for generating nonrecurrent rearrangements associated with genomic disorders. *Cell* **131**, 1235-1247 (2007).
- [redacted for confidentiality]**
16. Mielko, Z. et al. UV irradiation remodels the specificity landscape of transcription factors. *Proc Natl Acad Sci U S A* **120**, e2217422120 (2023).
 17. Abascal, F. et al. Somatic mutation landscapes at single-molecule resolution. *Nature* **593**, 405-410 (2021).
 18. Moore, L. et al. The mutational landscape of human somatic and germline cells. *Nature* (2021).

19. Chiarle, R. et al. Genome-wide translocation sequencing reveals mechanisms of chromosome breaks and rearrangements in B cells. *Cell* **147**, 107-119 (2011).

Below, we provide detailed responses to the reviewers' comments on the revised version of our manuscript.

REVIEWERS' COMMENTS

Reviewer #1 (Remarks to the Author):

In the revised study, Murat et al. have performed various additional analyses prompted by our remarks and those of other reviewers. Overall, we think the central conclusion of the study is substantially better supported after these revisions:

- some possible confounding factors have been ruled out, most prominently binding of TFs and CTCF
- G4s were further supported as causal to a part of the mutagenesis, and the G4 features leading to higher mutagenicity were elucidated
- an important adjustment for local trinucleotide composition at and around origins was implemented
- the new A>C (T>G) signature in the multi-tissue "cluster 2" was shown to be operative across many individual cancer types
- origins found by SNS-seq outside of the original ini-seq2 detected set were shown to be (modestly) mutagenic as well; both clustered and unclustered origins were mutagenic
- the SBS43-like signature observed has some more support that it is not an artefact, on the basis of tissue-specificity and association with active origins rather than only the -G4 repetitive sequences

We thank the reviewer for recognising the new experiments included in the revised version of our manuscript.

There are two outstanding issues from our side. Importantly, these are, in our opinion, not central for the conclusions of the study. Therefore, we would leave it at the authors' discretion to decide if these should be addressed. Even if these issues are not addressed, the study would be sufficiently solid, and findings robust, in the current state. We would be enthusiastic about seeing it published.

1- The subtype analysis, performed originally only on pancreatic cancers (PACA), based on the gene expression patterns. For these findings to be shown to be generally valid across cancer types, additional somatic tissues would need to show similar trends. While it is fine to present this result as-is -- clearly stating it is currently limited to PACA as other cancer types were not studied -- we do draw the authors' attention to the Hartwig Medical Foundation cancer WGS dataset, which has transcriptomes for ~3k samples currently. This would allow better testing of the proliferative driver/replicative stress hypothesis, even if in future work.

Incorporating additional datasets would certainly broaden the scope of our work; however, it would require complete reanalysis of our current data alongside these new samples. While we agree that this could further reinforce our findings, we believe the experiments presented in our manuscript sufficiently support our conclusions. The relationship between mutagenesis and cell proliferation is indeed of significant interest and may form the basis of a separate project in the future.

2- The mutational signature analysis, which was suggested by both us and by Reviewer 2 to be better addressed with a more standard, factorization (e.g. NMF) based approach, rather than a simpler approach they took by clustering the tumors based on the mutational spectrum clustering. Overall, we think the current approach is sufficient, and perhaps more appropriate with low mutation counts (their genomic regions-of-interest are short, which limits the number of mutations available). We do not think the main conclusion about increased mutagenesis at frequently-used origins would change if NMF were employed. It is possible they would recover a cleaner signature and/or multiple signatures, though; again as above we leave it to authors discretion to implement this change, or to leave it for future work.

We appreciate that using an NMF-based approach to detect and characterise the mutational processes at origins would align more closely with standard methodologies. We did explore this approach but found that

our chosen clustering method enabled more effective grouping of cancer samples, facilitating a clearer dissection of the various processes reported in the manuscript.

~~

Additionally, we provide our thoughts on the comments by Reviewer 2 below, as prompted by the editor. In our opinion, some general concerns of Reviewer 2 were not necessarily justified:

1- the reviewer 2 stated "work is undermined by well-understood alternative explanations that are ignored but lie at the bases of their analyses" but did not state clearly what these alternative explanations were.

It is possible they meant the (later mentioned) "differences in expression and a permissive chromatin environment impact DNA damage and repair associated with various mutational processes that are highly likely to coincide positionally with stable origins". Differences in expression (gene-resolution, 1-10kb) cannot explain the narrow peaks in mutagenesis (roughly 100 nt-resolution). Permissive chromatin environment as a factor seems to be ruled out by the new analysis that binding of certain TF does not explain the mutagenesis at origins. This seems to not be a concern.

2- the reviewer 2 stated "their analyses are flawed, relying upon poor statistical assertions" but apparently, to our understanding, did not show where the statistical analyses were flawed.

This may have referred to uses of the K-S or Mann-Whitney test, which is in our opinion not a substantial enough difference to dismiss the result.

This may also have referred to "all of these clusters are driven by a small number of origins", a comment addressed adequately by Rebuttal figure 9, which provides transparency on the number of origins. In summary, we do not share these worries about the study.

We thank the reviewer for assessing reviewer 2 concerns and for supporting our observations.

Some specific concerns of Reviewer 2 were valid critique. In our opinion, they were answered adequately by the authors of the study in their rebuttal. In specific:

3- Usage of one specific set of "constitutive origins" (iniseq2 derived) yet not other sets available in the literature (e.g. Nieduszynski paper). We would agree with the responses of the authors that various methods to identify origins such as the recent Nieduszynski methods, or OK-Seq or Bubble-seq would have either an insufficient power (former) or too low resolution (latter 2 methods) to be useful for their analysis of focal mutagenesis at origins. Therefore, it does not make too much sense to consult them here.

The one origin identification method that is useful for their analysis is SNS-seq; indeed in the revision they do measure mutation enrichment at SNS-seq peaks that were not already in their iniseq2 origin dataset, finding a modest enrichment of mutation rates. This supports their general claim that origins are mutagenic, and additionally supports that the less-used origins (in SNS but not already in iniseq2) are the less-mutagenized ones.

Overall, we think the authors did their "due diligence" regarding replication origin data sets considered, and that the iniseq2 set of origins they employ would be sufficient to support their main hypothesis.

4- Regarding analysis of mutation spectra. Reviewer 2 mentioned "the way you use mutation spectra, which has created a biased analysis" and "the lack of resemblance to extant signatures validates your approach to looking for mutagenic processes at origins, but the set up is biased in a way that will fail to resemble known signatures" and "multiple signatures packed on top of each other, consisting of multiple mutational processes, will generate weird signatures, even if they're across the same sorts of genomic feature. To do this properly in the future, you should consider de novo signature deconvolution and fitting."

In this remark, we would agree with the reviewer 2 in principle; we did also ask a related question (question 5 in the original review) as why the more standard NMF or similar factorization method was not applied, as

opposed to simply clustering the genomes of the spectra. The authors declined to implement this change. Upon consideration, we think implementing this change (while it would bring the study more in line with standard methodologies) is not crucial for the conclusions of the paper to stand. The clustering approach on mutational spectra, even though less powerful than NMF, does require fewer data points to work well and so might be advantageous for very narrow regions, such as the replication origins they analyze herein.

See the comments on the use of NMF-based approach above.

Reviewer #3 (Remarks to the Author):

The reviewer acknowledges that the authors have addressed most the concerns raised. The revised manuscript now includes additional analyses that strengthen the authors' arguments, particularly regarding mutation signatures at replication origins in non-proliferating and pre-cancerous tissues. The reviewer appreciates this effort, as it is crucial to establish that these are indeed cancer-specific events.

The reviewer understands that certain points could not be fully addressed due to the limitations caused by the dependence on existing high-quality datasets. As these challenges are beyond the scope of the manuscript and a single lab, they should not be viewed as a weakness.

The inclusion of new data from publicly available datasets, as well as the discussion on how the authors' work diverges from previous studies, adds substantial value to the manuscript. The manuscript should be accepted for publication.

We thank the reviewer for their suggestions, particularly the recommendation to analyse the mutational burden at origins in non-replicative and non-cancerous tissues. We believe these analyses enhance our manuscript by demonstrating that mutagenesis at origins is replication-dependent and intensified in cancers.

The reviewer found some minor issues in the Github page, which should be further clarify to be usable and produce reproducible results.

Reviewer #3 (Remarks on code availability):

The link is now available, but it is missing important information, particularly the R version and package dependencies. The data source is not indicated anywhere. Some links are broken (e.g., https://dcc.icgc.org/releases/release_28 [dcc.icgc.org]). Additionally, the link to the reference genomes is missing. The code also needs cleaning, and paths to local datasets should be renamed to something more meaningful and understandable for users.

Notably, the ICGC Data Portal was officially closed in June 2024, which accounts for some broken links. Although the interactive web portal has been decommissioned, the latest data release is still accessible via an SFTP server. Interested readers can refer to the Legacy ICGC 25K Data documentation for access instructions (<https://docs.icgc-argo.org/docs/data-access/icgc-25k-data#accessing-icgc-25k-release-data>). We have added a note in our code regarding data access. In this work, we do not use or reference a genome, so we did not provide a link. Finally, the file paths in the code are configured based on the computer where the analysis was conducted, allowing the code to be knitted into clean, easily readable HTML versions, which are also available on GitHub.

We did not perform test run for these codes, my lab is lacking capacity to do so at the moment.

Reviewer #4 (Remarks to the Author):
